# Multi-view Data Visualisation via Manifold Learning

## Abstract

Non-linear dimensionality reduction can be performed by *manifold learning* approaches, such as Stochastic Neighbour Embedding (SNE), Locally Linear Embedding (LLE) and Isometric Feature Mapping (ISOMAP). These methods aim to produce two or three latent embeddings, primarily to visualise the data in intelligible representations. This manuscript proposes extensions of Student's t-distributed SNE (t-SNE), LLE and ISOMAP, for dimensionality reduction and visualisation of *multi-view* data. Multi-view data refers to multiple types of data generated from the same samples.

The proposed multi-view approaches provide more comprehensible projections of the samples compared to the ones obtained by visualising each data-view separately. Commonly, visualisation is used for identifying underlying patterns within the samples. By incorporating the obtained low-dimensional embeddings from the multi-view manifold approaches into the $K$-means clustering algorithm, it is shown that clusters of the samples are accurately identified. Through extensive comparisons of novel and existing multi-view manifold learning algorithms on real and synthetic data, the proposed multi-view extension of t-SNE, named *multi-SNE*, is found to have the best performance. We further illustrate the applicability of the multi-SNE approach for the analysis of multi-omics single-cell data, where the aim is to visualise and identify cell heterogeneity and cell types in biological tissues relevant to health and disease.

## 1 Introduction

Data visualisation is an important and useful component of exploratory data analysis, as it can reveal interesting patterns in the data and potential clusters of the observations. A common approach for visualising high-dimensional data (data with a higher number of features ($p$) than samples ($n$), *i.e.* $p \gg n$) is by reducing its dimensions. Linear dimensionality reduction methods, including Principal Component Analysis (PCA) (Jolliffe & Cadima, 2016) and Non-negative Matrix Factorization (NMF) (García et al., 2018), assume linearity within data sets and as a result these methods often fail to produce reliable representations when linearity does not hold. *Manifold learning*, an active research area within machine learning, in contrast to the linear dimensionality reduction approaches do not rely on any linearity assumptions. By assuming that the dimensions of the data sets are artificially high, manifold learning methods aim to capture important information with minimal noise, in an induced low-dimensional embedding (Zheng & Xue, 2009). The generated low-dimensional embeddings can be used for data visualisation in the 2-D or 3-D spaces.

Manifold learning approaches used for dimensionality reduction and visualisation, focus on preserving at least one of the characteristics of the data. For example, the Stochastic Neighbour Embedding (SNE) preserves the probability distribution of the data (Hinton & Roweis, 2003). The Locally Linear Embedding (LLE) proposed by Roweis & Saul (2000) is a neighbourhood-preserving method. The Isometric Feature Mapping (ISOMAP) proposed by Tenenbaum et al. (2000) is a quasi-isometric method based on Multi-Dimensional Scaling (Kruskal, 1964). Spectral Embedding finds low-dimensional embeddings via spectral decomposition of the Laplacian matrix (Ng et al., 2001). The Local Tangent Space Alignment method proposed by Zhang & Zha (2004) learns the embedding by optimising local tangent spaces, which represent the local geometry of each neighbourhood, and Uniform Manifold Approximation and Projection (UMAP) preserves the global structure of the data by constructing a theoretical framework based on Riemannian geometry and algebraic topology (McInnes et al., 2018).

This manuscript focuses on data visualisation of *multi-view data*, which are regarded as different types of data sets that are generated on the same samples of a study. It is very common nowadays in many different fields to generate multiple data-views on the same samples. For example, multi-view imaging data describe distinct visual features such as local binary patterns (LBP), and histogram of oriented gradients (HOG) (Shen et al., 2013), while multi-omics data, *e.g. proteomics, genomics, etc*, in biomedical studies quantify different aspects of an organism's biological processes (Hasin et al., 2017). Through the collection of multi-view data, researchers are interested in better understanding the collected samples, including their visualisation, clustering and classification. Analysing simultaneously the multi-view data is not a straightforward task, as each data-view has its own distribution and variation pattern (Rodosthenous et al., 2020).

Several approaches have been proposed for the analysis of multi-view data. These include methods on clustering (Kumar et al., 2011; Liu et al., 2013; Sun et al., 2015; Ou et al., 2016; Ye et al., 2018; Ou et al., 2018; Wang & Allen, 2021), classification (Shu et al., 2019), regression (Li et al., 2019), integration (Rodosthenous et al., 2020) and dimensionality reduction (Sun, 2013; Zhao et al., 2018; Xu et al., 2015). Such approaches have been extensively discussed in the review papers of Xu et al. (2013) and Zhao et al. (2017).

In this manuscript, we focus on the visualisation task. By visualising multi-view data collectively the aim is to obtain a global overview of the data and identify patterns that would have potentially be missed if each data-view was visualised separately. Typically, multiple visualisations are produced, one from each data-view, or the features of the data-views are concatenated to produce a single visualisation. The former could provide misleading outcomes, with each data-view revealing different visualisations and patterns. The different statistical properties, physical interpretation, noise and heterogeneity between data-views suggest that concatenating features would often fail in achieving a reliable interpretation and visualisation of the data (Fu et al., 2008).

A number of multi-view visualisation approaches have been proposed in the literature, with some of these approaches based on the manifold approaches t-SNE and LLE. For example, Xie et al. (2011) proposed m-SNE that combines the probability distributions produced by each data-view into a single distribution via a weight parameter. The algorithm then implements t-SNE on the combined distribution to obtain a single low-dimensional embedding. The proposed solution finds the optimal choice for both the low-dimensional embeddings and the weight parameter simultaneously. Similarly, Kanaan Izquierdo (2017) proposed two alternative solutions based on t-SNE, named MV-tSNE1 and MV-tSNE2. MV-tSNE2 is similar to m-SNE combining the probability distributions through expert opinion pooling.

In parallel to our work, Hoan Do & Canzar (2021) proposed a multi-view extension of t-SNE, named j-SNE. Both multi-SNE and j-SNE firstly appeared as preprints in January 2021[1]. J-SNE produces low-dimensional embeddings through an iterative procedure that assigns each data-view a weight value that is updated per iteration through regularisation.

In addition, Shen et al. (2013) proposed multi-view Locally Linear Embeddings (m-LLE) that is an extension of LLE for effectively retrieving medical images. M-LLE produces a single low-dimensional embedding by integrating the embeddings from each data-view according to a weight parameter $c$, which refers to the contribution of each data-view. Similarly to m-SNE, the algorithm optimizes both the weight parameter and the embeddings simultaneously. Zong et al. (2017) proposed MV-LLE that minimises the cost function by assuming a consensus matrix across all data-views.

Building on the existing literature work, we propose here alternative extensions to the manifold approaches: t-SNE, LLE, and ISOMAP, for visualising multi-view data. The cost functions of our proposals are different from the existing ones, as they integrate the available information from the multi-view data iteratively. At each iteration, the proposed *multi-SNE* updates the low-dimensional embeddings by minimising the dissimilarity between their probability distribution and the distribution of each data-view. The total cost of this approach equals to the weighted sum of those dissimilarities. Our proposed variation of LLE, *Multi-LLE*, constructs the low-dimensional embeddings by utilising a consensus weight matrix, which is taken as the weighted sum of the weight matrices computed by each data-view. Lastly, the low-dimensional embeddings

---

[1]https://doi.org/10.1101/2021.01.10.426098

in the proposed *multi-ISOMAP* are constructed by using a consensus graph, for which the nodes represent the samples and the edge lengths are taken as the averaged distance between the samples in each data-view. *M-ISOMAP* is proposed as an alternative ISOMAP-based multi-view manifold learning algorithm. Similar to m-SNE and m-LLE, m-ISOMAP provide a weighted integration of the low-dimensional embeddings produced by the implementation of ISOMAP on each data-view separately.

As the field of multi-view data analysis is relatively new, the literature lacks comparative studies between multi-view manifold learning algorithms. This manuscript makes a novel contribution to the field by conducting extensive comparisons between the multi-view non-linear dimensionality reduction approaches proposed in this manuscript, multi-SNE, multi-LLE, multi-ISOMAP and m-ISOMAP with other approaches proposed in the literature. These comparisons are conducted on both real and synthetic data that have been designed to capture different data characteristics. The aim of these comparisons is to identify the best-performing algorithms, discuss pitfalls of the approaches and guide the users to the most appropriate solution for their data.

We illustrate that our proposals result to more robust solutions compared to the approaches proposed in the literature, including m-SNE, m-LLE and MV-SNE. We further illustrate through the visualisation of the low-dimensional embeddings produced by the proposed multi-view manifold learning algorithms, that if clusters exist within the samples, they can be successfully identified. We show that this can be achieved by applying the $K$-means algorithm on the low-dimensional embeddings of the data. The $K$-means (MacQueen, 1967) was chosen to cluster the data points, as it is one of the most famous and prominent partition clustering algorithms (Xu & Tian, 2015). A better clustering performance by $K$-means suggests a visually clearer separation of clusters. Through the conducted experiments, we show that the proposed multi-SNE approach recovers well-separated clusters of the data, and has comparable performance to multi-view clustering algorithms that exist in the literature.

## 2 Material and Methods

In this section, the proposed approaches for multi-view manifold learning are described. This section starts with an introduction of the notation used throughout this manuscript. The proposed multi-SNE, multi-LLE and multi-ISOMAP are described in Sections 2.2, 2.3 and 2.4, respectively. The section ends with a description of the process for tuning the parameters of the algorithms.

### 2.1 Notation

Throughout this paper, the following notation is used:

- $N$: The number of samples.

- $X \in \mathbb{R}^{N \times p}$: A single-view data matrix, representing the original high-dimensional data used as input; $\mathbf{x}_i \in \mathbb{R}^p$ is the $i^{th}$ data point of $X$.

- $M$: The number of data-views in a given data set; $m \in \{1, \cdots, M\}$ represents an arbitrary data-view.

- $X^{(m)} \in \mathbb{R}^{N \times p_m}$: The $m^{th}$ data-view of multi-view data; $\mathbf{x}_i^m \in \mathbb{R}^{p_m}$ is the $i^{th}$ data point of $X^{(m)}$.

- $Y \in \mathbb{R}^{N \times d}$: A low-dimensional embedding of the original data. $\mathbf{y}_i \in \mathbb{R}^d$ represents the $i^{th}$ data point of $Y$. In this manuscript, $d = 2$, as the focus of the manuscript is on data visualisation.

### 2.2 Multi-SNE

SNE, proposed by Hinton & Roweis (2003), measures the probability distribution, $P$ of each data point $\mathbf{x}_i$ by looking at the similarities among its neighbours. For every sample $i$ in the data, $j$ is taken as its potential neighbour with probability $p_{ij}$, given by

$$p_{ij} = \frac{\exp\left(-d_{ij}^2\right)}{\sum_{k \neq i} \exp\left(-d_{ik}^2\right)}, \tag{1}$$

where $d_{ij} = \frac{||\mathbf{x}_i - \mathbf{x}_j||^2}{2\sigma_i^2}$ represents the dissimilarity between points $\mathbf{x}_i$ and $\mathbf{x}_j$. The value of $\sigma_i$ is either set by hand or found by binary search (van der Maaten & Hinton, 2008). Based on this value, a probability distribution of sample $i$, $P_i = \sum_j p_{ij}$, with fixed perplexity is produced. Perplexity refers to the effective number of local neighbours and it is defined as $Perp(P_i) = 2^{H(P_i)}$, where $H(P_i) = -\sum_j p_{ij} \log_2 p_{ij}$ is the Shannon entropy of $P_i$. It increases monotonically with the variance $\sigma_i$ and typically takes values between 5 and 50.

In the same way, a probability distribution in the low-dimensional space, $Y$, is computed as follows:

$$q_{ij} = \frac{\exp\left(-||\mathbf{y}_i - \mathbf{y}_j||^2\right)}{\sum_{k \neq i} \exp\left(-||\mathbf{y}_i - \mathbf{y}_k||^2\right)}, \tag{2}$$

which represents the probability of point $i$ selecting point $j$ as its neighbour.

The induced embedding output, $\mathbf{y}_i$, represented by probability distribution, $Q$, is obtained by minimising the Kullback-Leibler divergence (KL-divergence) $KL(P||Q)$ between the two distributions $P$ and $Q$ (Kullback & Leibler, 1951). The aim is to minimise the cost function:

$$C_{SNE} = \sum_i KL(P_i||Q_i) = \sum_i \sum_j p_{ij} \log \frac{p_{ij}}{q_{ij}} \tag{3}$$

Hinton & Roweis (2003) assumed a Gaussian distribution in computing the similarity between two points in both high and low dimensional spaces. van der Maaten & Hinton (2008) proposed a variant of SNE, called t-SNE, which uses a symmetric version of SNE and a Student t-distribution to compute the similarity between two points in the low-dimensional space $Q$, given by

$$q_{ij} = \frac{(1 + ||\mathbf{y}_i - \mathbf{y}_j||^2)^{-1}}{\sum_{k \neq l}(1 + ||\mathbf{y}_k - \mathbf{y}_l||^2)^{-1}} \tag{4}$$

T-SNE is often preferred, because it reduces the effect of crowding problem (limited area to accommodate all data points and differentiate clusters) and it is easier to optimise, as it provides simpler gradients than SNE (van der Maaten & Hinton, 2008).

We propose *multi-SNE*, a multi-view manifold learning algorithm based on t-SNE. Our proposal computes the KL-divergence between the distribution of a single low-dimensional embedding and each data-view of the data separately, and minimises their weighted sum. An iterative algorithm is proposed, in which at each iteration the induced embedding is updated by minimising the cost function:

$$C_{multi-SNE} = \sum_m \sum_i \sum_j w^m p_{ij}^m \log \frac{p_{ij}^m}{q_{ij}}, \tag{5}$$

where $w^m$ is the combination coefficient of the $m^{th}$ data-view. The vector $\mathbf{w} = (w^1, \cdots, w^M)$ acts as a weight vector that satisfies $\sum_m w^m = 1$. In this study, equal weights on all data-views were considered, *i.e.* $w^m = \frac{1}{M}, \quad \forall m = 1, \cdots, M$. The algorithm of the proposed multi-SNE approach is presented in Appendix A.

An alternative multi-view extension of t-SNE, called *m-SNE* was proposed by Xie et al. (2011). M-SNE applies t-SNE on a single distribution in the high-dimensional space, which is computed by combining the probability distributions of the data-views, given by $p_{ij} = \sum_{m=1}^M \beta^m p_{ij}^m$. The coefficients (or weights) $\beta^m$ share the same role as $w^m$ in multi-SNE and similarly $\boldsymbol{\beta} = (\beta^1, \cdots, \beta^M)$ satisfies $\sum_m \beta^m = 1$. This leads to a different cost function than the one in equation (5).

Kanaan Izquierdo (2017) proposed a similar cost function for multi-view t-SNE, given as follows:

$$C_{\text{MV-tSNE1}} = \sum_m \sum_i \sum_j p_{i|j}^m \log \frac{p_{i|j}^m}{q_{i|j}} \tag{6}$$

Their proposal is a special case of multi-SNE, with $w_m = \frac{1}{M}$. Kanaan Izquierdo (2017) did not pursue MV-tSNE1 any further, but instead, they proceeded with an alternative solution, MV-tSNE2, which combines the probability distributions (similar to m-SNE) through expert opinion pooling. A comparison between multi-SNE and MV-tSNE2 is presented in Appendix D.1. Based on two real data sets, multi-SNE and m-SNE outperformed MV-tSNE2, with the solution by multi-SNE producing the best separation among the clusters in both examples.

Multi-SNE avoids combining the probability distributions of all data-views together. Instead, the induced embeddings are updated by minimising the KL-divergence between every data-view's probability distribution and that of the low-dimensional representation we seek to obtain. In other words, this is achieved by computing and summing together the gradient descent for each data-view. The induced embedding is then updated by minimising the summed gradient descent.

Throughout this paper, for all variations of t-SNE we have applied the PCA pre-training step proposed by van der Maaten & Hinton (2008). van der Maaten & Hinton (2008) discussed that by reducing the dimensions of the input data through PCA the computational time of t-SNE is reduced. In this paper, the principal components taken retained at least 80% of the total variation (variance explained) in the original data. In addition, as the multi-SNE algorithm is an iterative algorithm we opted for running the algorithm for 1,000 iterations for all analyses conducted. Alternatively, a stopping rule could have been implemented with the iterative algorithm to stop after no significant changes were observed to the cost-function. Both these options are available at the implementation of the multi-SNE algorithm.

The original t-SNE implementation was applied in the presented work. All t-SNE results presented in this manuscript were based on the original `R` implementation (https://cran.r-project.org/web/packages/tsne/) and verified by the original `Python` implementation (`https://lvdmaaten.github.io/tsne/`).

## 2.3   Multi-LLE

LLE attempts to discover a non-linear structure of high-dimensional data, $X$, by computing low-dimensional and neighbourhood-preserving embeddings, $Y$ (Saul & Roweis, 2001). The main three steps of the algorithm are:

1. The set, denoted by $\Gamma_i$, contains the $K$ nearest neighbours of each data point $\mathbf{x}_i, i = 1, \cdots, N$. The most common distance measure between the data points is the Euclidean distance. Other local metrics can also be used in identifying the nearest neighbours (Roweis & Saul, 2000).

2. A weight matrix, $W$, is computed, which acts as a bridge between the high-dimensional space in $X$ and the low-dimensional space in $Y$. Initially, $W$ reconstructs $X$, by minimising the cost function:

$$\mathcal{E}_X = \sum_i |\mathbf{x}_i - \sum_j W_{ij}\mathbf{x}_j|^2 \tag{7}$$

   where the weights $W_{ij}$ describe the contribution of the $j^{th}$ data point to the $i^{th}$ reconstruction. The optimal weights $W_{ij}$ are found by solving the least squares problem given in equation (7) subject to the constraints:

   (a) $W_{ij} = 0$, if $j \notin \Gamma_i$, and
   (b) $\sum_j W_{ij} = 1$

3. Once $W$ is computed, the low-dimensional embedding $\mathbf{y}_i$ of each data point $i = 1, \cdots, N$, is obtained by minimising:

$$\mathcal{E}_Y = \sum_i |\mathbf{y}_i - \sum_j W_{ij}\mathbf{y}_j|^2 \tag{8}$$

   The solution to equation (8), is obtained by taking the bottom $d$ non-zero eigenvectors of the sparse $N \times N$ matrix, $M = (I - W)^T (I - W)$ (Roweis & Saul, 2000).

We propose *multi-LLE*, a multi-view extension of LLE, that computes the low-dimensional embeddings by using the consensus weight matrix:

$$\hat{W} = \sum_m \alpha^m W^m \tag{9}$$

where $\sum_m \alpha^m = 1$, and $W^m$ is the weight matrix for each data-view $m = 1, \cdots M$. Thus, $\hat{Y}$ is obtained by solving:

$$\mathcal{E}_{\hat{Y}} = \sum_i |\hat{\mathbf{y}}_i - \sum_j \hat{W}_{ij} \hat{\mathbf{y}}_j|^2$$

The multi-LLE algorithm is presented in Appendix A.

Shen et al. (2013) proposed *m-LLE*, an alternative multi-view extension of LLE. The LLE embeddings of each data-view are combined and LLE is applied to each data-view separately. The weighted average of those embeddings is taken as the unified low-dimensional embedding. In other words, computing the weight matrices $W^m$ and solving $\mathcal{E}_{Y^m} = \sum_i |\mathbf{y}_i^m - \sum_j W_{ij}^m \mathbf{y}_j^m|^2$, for each $m = 1, \cdots M$ separately. Thus, the low-dimensional embedding $\hat{Y}$ is computed by $\hat{Y} = \sum_m \beta^m Y^m$, where $\sum_m \beta^m = 1$.

An alternative multi-view LLE solution was proposed by Zong et al. (2017) to find a consensus manifold, which is then used for multi-view clustering via Non-negative Matrix Factorization; we refer to this approach as *MV-LLE*. This solution minimises the cost function by assuming a consensus weight matrix across all data-views, as given in equation (9). The optimisation is then solved by using the Entropic Mirror Descent Algorithm (EMDA) (Beck & Teboulle, 2003). In contrast to m-LLE and MV-LLE, multi-LLE combines the weight matrices obtained from each data-view, instead of the LLE embeddings. No comparisons were conducted with MV-LLE and the proposed multi-LLE, as the code of the MV-LLE algorithm is not publicly available.

## 2.4 Multi-ISOMAP

ISOMAP aims to discover a low-dimensional embedding of high-dimensional data by maintaining the geodesic distances between all points (Tenenbaum et al., 2000); it is often regarded as an extension of Multi-dimensional Scaling (MDS) (Kruskal, 1964). The ISOMAP algorithm comprises of the following three steps:

Step 1. **A graph is defined.** Let $G \sim (V, E)$ define a neighbourhood graph, with vertices $V$ representing all data points. The edge length between any two vertices $i, j \in V$ is defined by the distance metric $d_X(i, j)$, measured by the Euclidean distance. If a vertex $j$ does not belong to the $K$ nearest neighbours of $i$, then $d_X(i, j) = \infty$. The parameter $K$ is given as input, and it represents the connectedness of the graph $G$; as $K$ increases, more vertices are connected.

Step 2. **The shortest paths between all pairs of points in $G$ are computed.** The shortest path between vertices $i, j \in V$ is defined by $d_G(i, j)$. Let $D_G \in \mathbb{R}^{|V| \times |V|}$ be a matrix containing the shortest paths between any vertices $i, j \in V$, defined by $(D_G)_{ij} = d_G(i, j)$.

The most efficient known algorithm to perform this task is Dijkstra's Algorithm (Dijkstra, 1959). In large graphs, an alternative approach to Dijkstra's Algorithm would be to initialize $d_G(i, j) = d_X(i, j)$ and replace all entries by $d_G(i, j) = \min \{d_G(i, k), d_G(k, j)\}$.

Step 3. **The low-dimensional embeddings are constructed.** The $i^{th}$ component of the low-dimensional embedding is given by $y_i = \sqrt{\lambda_p} u_p^i$, where $u_p^i$ the $i^{th}$ component of $p^{th}$ eigenvector and $\lambda_p$ is the $p^{th}$ eigenvalue in decreasing order of the the matrix $\tau(D_G)$ (**?**). The operator, $\tau$ is defined by $\tau(D) = -\frac{HSH}{2}$, where $S$ is the matrix of squared distances defined by $S_{ij} = D_{ij}^2$, and $H$ is defined by $H_{ij} = \delta_{ij} - \frac{1}{N}$. This is equivalent to applying classical MDS to $D_G$, leading to a low-dimensional embedding that best preserves the manifold's estimated intrinsic geometry.

Multi-ISOMAP is our proposal for adapting ISOMAP on multi-view data. Let $G_m \sim (V, E_m)$ be a neighbourhood graph obtained from data-view $X^{(m)}$ as defined in the first step of ISOMAP. All neighbourhood

graphs are then combined into a single graph, $\tilde{G}$; the combination is achieved by computing the edge length as the averaged distance of each data-view, *i.e.* $d_{\tilde{G}}(i, j) = w_m \sum_m d_{G_m}(i, j)$. Once a combined neighbourhood graph is computed, multi-ISOMAP follows steps 2 and 3 of ISOMAP described above. For simplicity, the weights throughout this paper were set as $w_m = \frac{1}{M}, \forall m$. The multi-ISOMAP algorithm is presented in Appendix A.

For completion, we have in addition adapted ISOMAP for multi-view visualisation following the framework of both m-SNE and m-LLE. Following the same logic, *m-ISOMAP* combines the ISOMAP embeddings of each data-view by taking the weighted average of those embeddings as the unified low-dimensional embedding. In other words, the low-dimensional embedding $\hat{Y}$ is obtained by computing $\hat{Y} = \sum_m \beta^m Y^m$, where $\sum_m \beta^m = 1$.

### 2.5 Parameter Tuning

The multi-view manifold learning algorithms were tested on real and synthetic data sets for which the samples can be separated into several clusters. The true clusters are known and they were used to tune the parameters of the methods. To quantify the clustering performance, we used the following four extrinsic measures: (i) Accuracy (ACC), (ii) Normalised Mutual Information (NMI) (Vinh et al., 2010), (iii) Rand Index (RI) (Rand, 1971) and (iv) Adjusted Rand Index (ARI) (Hubert & Arabie, 1985). All measures take values in the range $[0, 1]$, with 0 expressing complete randomness, and 1 perfect separation between clusters. The mathematical formulas of the four measures are presented in Appendix B.

SNE, LLE and ISOMAP depend on parameters of which their proper tuning ensues to optimal results. LLE and ISOMAP depend on the number of nearest neighbours ($NN$). SNE depends on the Perplexity ($Perp$) parameter, which is directly related to the number of nearest neighbours. Similarly, the multi-view extensions of the three methods depend on the same parameters. The choice of the parameter can influence the visualisations and in some cases present the data into separate maps (van der Maaten & Hinton, 2012).

By assuming that the data samples belong to a number of clusters that we seek to identify, the performance of the algorithms was measured for a range of tuning parameter values, $S = \{2, 10, 20, 50, 80, 100, 200\}$. Note that for all algorithms, the parameter value cannot exceed the total number of samples in the data.

For all manifold learning approaches, the following procedure was implemented to tune the optimal parameters of each method per data set:

1. The method was applied for all different parameter values in $S$.

2. The $K$-means algorithm was applied to the low-dimensional embeddings produced for each parameter value

3. The performance of the chosen method was evaluated quantitatively by computing ACC, NMI, RI and ARI for all tested parameter values.

The optimal parameter value was finally selected based on the evaluation measures. Section 4.3 explores how the different approaches are affected by their parameter values. For the other subsections of Section 4, the optimal parameter choice per approach was used for the comparison of the multi-view approaches. Section 4.3 presents the process of parameter tuning on the synthetic data analysed, and measured the performance of single-view and multi-view manifold learning algorithms. The same process was repeated for the real data analysed (see Appendix D.3 for more information).

## 3   Data

Data sets with different characteristics were analysed to explore and compare the proposed multi-view manifold learning algorithms under different scenarios (Table 1). The methods were evaluated on data sets that have a different number of data-views, clusters and sample sizes. The real data sets analysed are classified as heterogeneous, due to the nature of their data, while the synthetic data sets are classified

as non-heterogeneous, since they were generated under the same conditions and distributions. Both high-dimensional ($p \gg N$) and low-dimensional data sets were analysed. Through these comparisons, we wanted to investigate how the multi-view methods perform and how they compare with single-view methods.

In this section, we describe the synthetic and real data sets analysed in the manuscript. Some of the real data sets analysed have previously been used in the literature for examining different multi-view algorithms, for example, data integration (Wang et al., 2014) and clustering (Ou et al., 2018).

## 3.1 Synthetic Data

A motivational multi-view example was constructed to qualitatively evaluate the performance of multi-view manifold learning algorithms against their corresponding single-view algorithms. Its framework was designed specifically to produce distinct projections of the samples from each data-view. Additional synthetic data sets were generated to explore how the algorithms behave when the separation between the clusters exists, but it is not as explicit as in the motivational example.

All synthetic data were generated using the following process. For the same set of samples, a specified number of data-views were generated, with each data-view capturing different information of the samples. Each data-view, $m$ follows a multivariate normal distribution with mean vector $\boldsymbol{\mu_m} = (\mu_1, \cdots, \mu_{p_m})^T$ and covariance matrix $\Sigma_m = I_{p_m}$, where $p_m$ is the number of features in the $m^{th}$ data-view. The matrix $I_{p_m}$ represents a $p_m \times p_m$ identity matrix. For each data-view, different $\boldsymbol{\mu_m}$ values were chosen to distinguish the clusters. Noise, $\boldsymbol{\epsilon}$, following a multivariate normal distribution with mean $\boldsymbol{\mu_\epsilon} = 0$ and covariance matrix $\Sigma_\epsilon = I_{p_m}$ was added to increase randomness within each data-view. Noise, $\boldsymbol{\epsilon}$, increases the variability within a given data-view. The purpose of this additional variability is to assess whether the algorithms are able to equally capture information from all data-views and are not biased towards the data-view(s) with a higher variability. Thus, noise, $\boldsymbol{\epsilon}$, was only included in selected data-views and not in the rest. Although this strategy is equivalent to sampling once using a larger variance, the extra noise explicitly distinguishes the data-view with the higher variability from the rest.

In other words, $X \sim MVN(\boldsymbol{\mu_m}, \Sigma_m) + \boldsymbol{\epsilon}$, where $MVN$ represents multivariate normal distribution. Distinct polynomial functions (*e.g.* $h(x) = x^4 + 3x^2 + 5$) were randomly generated for each data-view and applied on the samples to express non-linearity. The last step was performed to ensure that linear dimensionality reduction methods (*e.g.* PCA) would not successfully cluster the data.

The three synthetic data sets with their characteristics are described next.

### 3.1.1 Motivational Multi-view Data Scenario (MMDS)

Assume that the truth underlying structure of the data separates the samples into three true clusters as presented in Figure 1. Each synthetic data-view describes the samples differently, which results in three distinct clusterings, none of which reflects the global underlying truth. In particular, the first view separates only cluster **C** from the others (View 1 in Figure 1), the second view separates only cluster **B** (View 2) and the third view separates only cluster **A** (View 3). In this scenario, only the third data-view contained an extra noise parameter, $\boldsymbol{\epsilon}$, resulting in a data-view with a higher variability than the other two data-views.

### 3.1.2 Noisy data-view scenario (NDS)

A synthetic data set which consists of 4 data-views and 3 true underlying clusters was generated. The first three data-views follow the same structure as MMDS, while the $4^{th}$ data-view represents a completely noisy data-view, *i.e.* with all data points lying in a single cluster. The rationale for creating such a data set is to examine the effect of the noisy data views in the multi-view visualisation and clustering. This data set was used to show that the multi-view approaches can identify not useful data-views and discard them. For $n = 300$ equally balanced data samples, the data-views contain $p_m = 100, \forall m = 1, 2, 3, 4$, features. To summarise, NDS adds a noisy data-view to the MMDS data set.

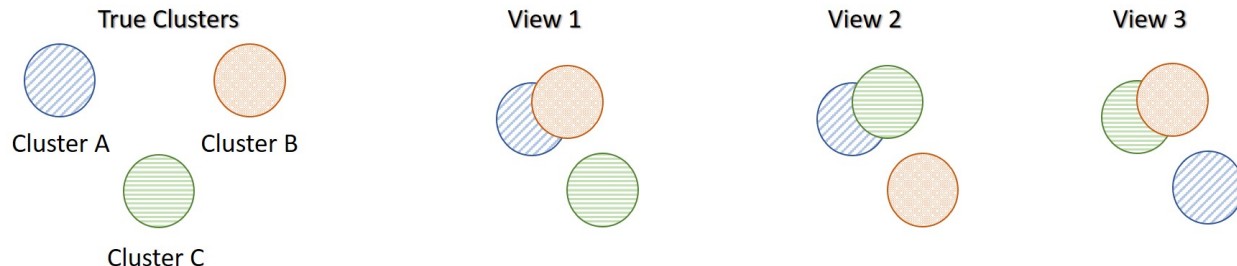

Figure 1: **Motivational Multi-view Data Scenario (MMDS).** Each data-view captures different characteristics of the three clusters, and thus produces different clusterings.

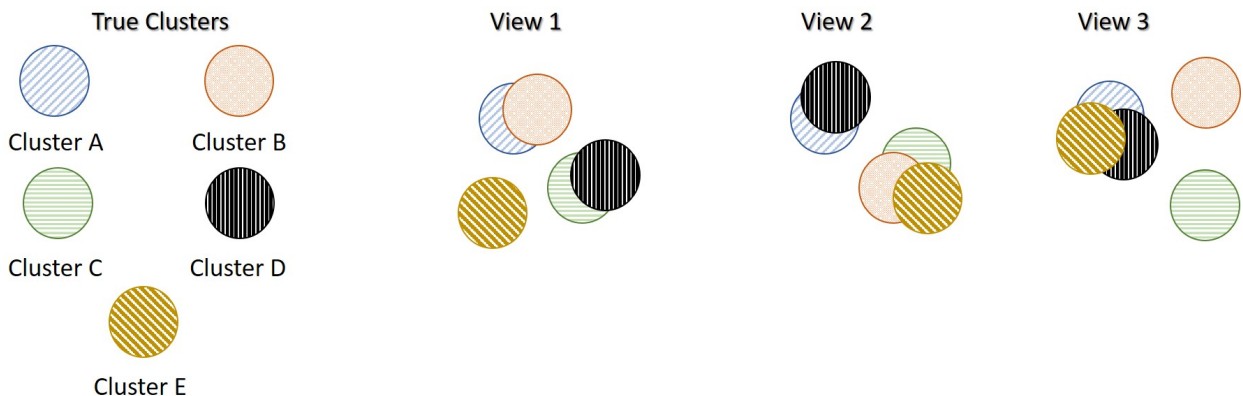

Figure 2: **More Clusters than data-views Scenario (MCS).** In this example, there are 3 data views but 5 true underlying clusters. Each data-view captures different characteristics of the five clusters, and thus produces different clusterings.

### 3.1.3 More clusters than data-views scenario (MCS)

A synthetic data set that was generated similarly to MMDS but with 5 true underlying clusters instead of 3. The true underlying structure of the each data-view is shown in Figure 2. In this data set, $p_v = 100, \forall v$ features were generated on $n = 500$, equally balanced data samples. In comparison with MMDS, MCS contains more clusters, but the same number of data-views. Similarly to MMDS and NDS, in this scenario, only the third data-view contained an extra noise parameter, $\epsilon$, resulting in a data-view with a higher variability than the other two data-views.

## 3.2 Real Data

The three real data sets analysed in the study are described below.

### 3.2.1 Cancer Types [2]

This data set includes *65* patients with breast cancer, *82* with kidney cancer and *106* with lung cancer. For each patient the three data-views are available: (a) genomics ($p_1 = 10299$ genes), (b) epigenomics ($p_2 = 22503$ methylation sites) and (c) transcriptomics ($p_3 = 302$ mi-RNA sequences). The aim is to cluster patients by their cancer type (Wang et al., 2014).

---

[2]http://compbio.cs.toronto.edu/SNF/SNF/Software.html

### 3.2.2 Caltech7 [3]

Caltech-101 contains pictures of objects belonging to 101 categories. This publicly available subset of Caltech-101 contains 7 classes. It consists of *1474* objects on six data-views: (a) Gabor ($p_1 = 48$), (b) wavelet moments ($p_2 = 40$), (c) CENTRIST ($p_3 = 254$), (d) histogram of oriented gradients ($p_4 = 1984$), (e) GIST ($p_5 = 512$), and (f) local binary patterns ($p_6 = 928$) (Fei-Fei et al., 2006).

### 3.2.3 Handwritten Digits [4]

This data set consists of features on handwritten numerals $(0-9)$ extracted from a collection of Dutch utility maps. Per class *200* patterns have been digitised in binary images (in total there are 2000 patterns). These digits are represented in terms of six data-views: (a) Fourier coefficients of the character shapes ($p_1 = 76$), (b) profile correlations ($p_2 = 216$), (c) Karhunen-Love coefficients ($p_3 = 64$), (d) pixel averages in 2 x 3 windows ($p_4 = 240$), (e) Zernike moments ($p_5 = 47$) and (f) morphological features ($p_6 = 6$) (Dua & Graff, 2017).

The handwritten digits data set is characterised by having perfectly balanced data samples; each of the ten clusters contains exactly 200 numerals. On the other hand, caltech7 is an imbalanced data set with the first two clusters containing many more samples than the other clusters. The number of samples in each cluster is {A: 435, B: 798, C: 52, D: 34, E: 35, F: 64, G: 56}. The performance of the methods was explored on both the imbalanced caltech7 data set and a balanced version of the data, for which 50 samples from clusters $A$ and $B$ were randomly selected.

## 4 Results

In this section, we illustrate the application and evaluation of the proposed multi-view extensions of t-SNE, LLE and ISOMAP on real and synthetic data. Comparisons between the multi-view solutions, along with their respective single-view solutions are implemented. A trivial solution is to concatenate the features of all data-views into a large single data matrix and apply on this dataset a single-view manifold learning algorithm. Since it is likely that each data-view has different variability, each data-view was firstly normalised before concatenation to ensure the same variability across all data-views. Normalisation was achieved by removing the mean and dividing by the standard deviation of the features in all data-views.

In the following subsections we have addressed the following:

1. **Can multi-view manifold learning approaches obtain better visualisations than single-view approaches?** The performance of the multi-view approaches in visualising the underlying structure of the data is illustrated. It is shown how the underlying structure is misrepresented when individual data sets or the concatenated data set are visualised.

2. **The visualisations of multi-view approaches are quantitatively evaluated using $K$-means**. By extracting the low dimensional embeddings of the multi-view approaches and inputting them as features in the clustering algorithm $K$-means, we have quantitatively evaluated the performance of the approaches for identifying underlying clusters and patterns within the data.

3. **The effect of the parameter values on the multi-view manifold learning approaches was explored.** As discussed the proposed multi-view manifold approaches depend on a parameter that requires tuning. In a series of experiments, we investigated the effect that the parameter value has on each approach. This was done by exploring both the visualisations produced and by evaluating the clustering of the approaches for different parameter values.

4. **Should we use all available data-views? If some data-views contain more noise than signal, should we discard them?** These are two crucial questions that concern every researcher

---

[3]https://github.com/yeqinglee/mvdata
[4]https://archive.ics.uci.edu/ml/datasets/Multiple+Features

**Data Description**

| | Data Set | Views $(M)$ | Clusters $(k)$ | Features $(p_{largest})$ | Samples $(N)$ | Hetero-geneous | High dimensional |
|---|---|---|---|---|---|---|---|
| | Cancer Types | 3 | 3 | 22503 | 253 | ✓ | ✓ |
| Real | Caltech7 | 6 | 7 | 1984 | 1474 | ✓ | ✓ |
| | Handwritten Digits | 6 | 10 | 240 | 2000 | ✓ | ✗ |
| | MMDS | 3 | 3 | 300 | 300 | ✗ | ✗ |
| Synthetic | NDS | 4 | 3 | 400 | 300 | ✗ | ✓ |
| | MCS | 3 | 5 | 300 | 500 | ✗ | ✗ |

Table 1: **The characteristics of the data sets analysed.** The number of views, number of clusters, the largest number of features amongst the data views, and the number of samples for both the real and synthetic data sets analysed are presented. Real data are taken as heterogeneous, whereas the synthetic data are regarded as homogeneous. High-dimensional data contain more features than samples $(p \gg N)$.

working with multi-view data; are all data-views necessary and beneficial to the final outcome? We have addressed these questions by analysing data sets that contain noisy data. By investigating both the produced visualisations and evaluating the clusterings obtained with and without the noisy data, we discuss why it is not always beneficial to include all available data views.

The section ends by proposing alternative variations for the best-performing approach, multi-SNE. Firstly, a proposal for automatically computing the weights assigned to each data-view. In addition, we explore an alternative pre-training step for multi-SNE, where instead of conducting PCA on each data-view, multi-CCA is applied on the multiple data-views for reducing their dimensions into a latent space of uncorrelated embeddings (Rodosthenous et al., 2020).

## 4.1 Comparison Between Single-view and Multi-view Visualisations

Visualising multi-view data can be trivially achieved either by looking at the visualisations produced by each data-view, or by concatenating all features into a long vector. T-SNE, LLE and ISOMAP applied on every single data-view of the MMDS data set separately capture the correct local underlying structure of the respective data-view (Figure 3). However, by design, they cannot capture the global structure of the data. **SNE**$_{concat}$, **LLE**$_{concat}$ and **ISOMAP**$_{concat}$ represent the trivial solutions of concatenating the features of all data-views before applying t-SNE, LLE and ISOMAP, respectively. These trivial solutions capture mostly the structure of the third data-view, because that data-view has a higher variability between the clusters than the other two.

Multi-SNE, multi-LLE and multi-ISOMAP produced the best visualisations out of all SNE-based, LLE-based and ISOMAP-based approaches, respectively. These solutions were able to separate clearly the three true clusters, with multi-SNE showing the clearest separation between them. Even though m-SNE separates the samples according to their corresponding clusters, this separation would not be recognisable if the true labels were unknown, as the clusters are not sufficiently separated. The visualisation by m-LLE was similar to the ones produced by single-view solutions on concatenated features, while m-ISOMAP bundles all samples into a single cluster.

By visualising the MMDS data set via both single-view and multi-view clustering approaches, multi-SNE has shown the most promising results (Figure 3). We have shown that single-view analyses may lead to conflicting results, while multi-view approaches are able to capture the true underlying structure of the synthetic MMDS.

## 4.2 Multi-view Manifold Learning for Clustering

It is very common in studies to utilise the visualisation of data to identify any underlying patterns or clusters within the data samples. Here, it is illustrated how the multi-view approaches can be used to identify such clusters. To quantify the visualisation of the data, we applied the $K$-means algorithm on the low-dimensional embeddings produced by multi-view manifold learning algorithms. If the two-dimensional

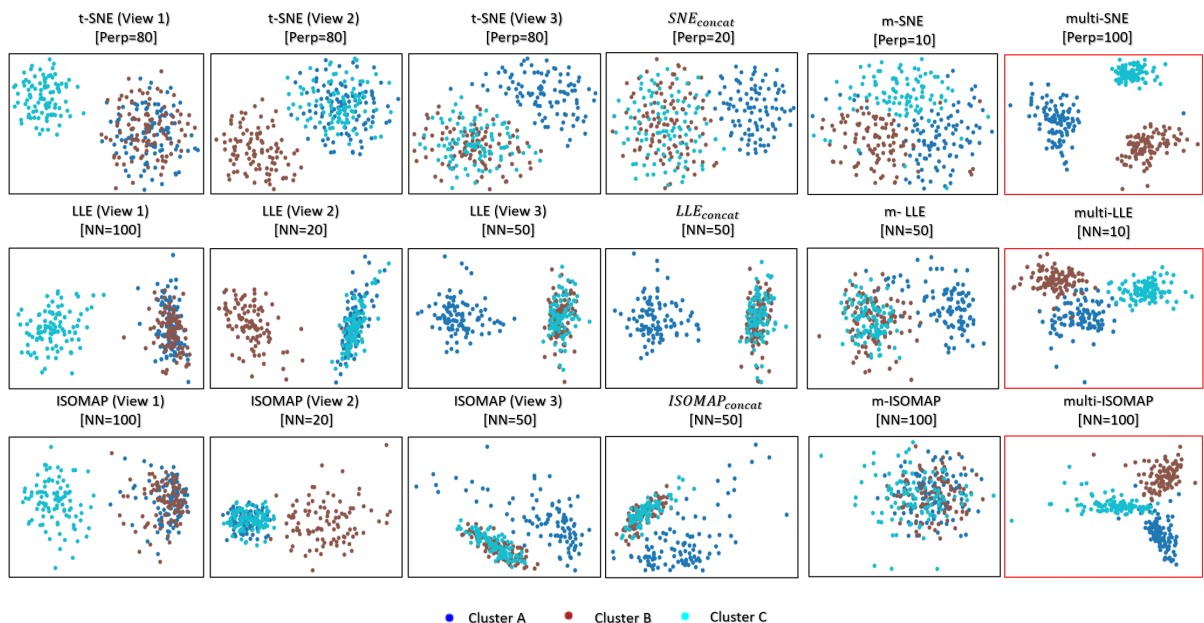

Figure 3: **Visualisations of MMDS.** Projections produced by the SNE, LLE, ISOMAP based algorithms. The projections within the red frame present our proposed methods: multi-SNE, multi-LLE and multi-ISOMAP. The parameters $Perp$ and $NN$ refer to the optimised perplexity and number of nearest neighbours, respectively.

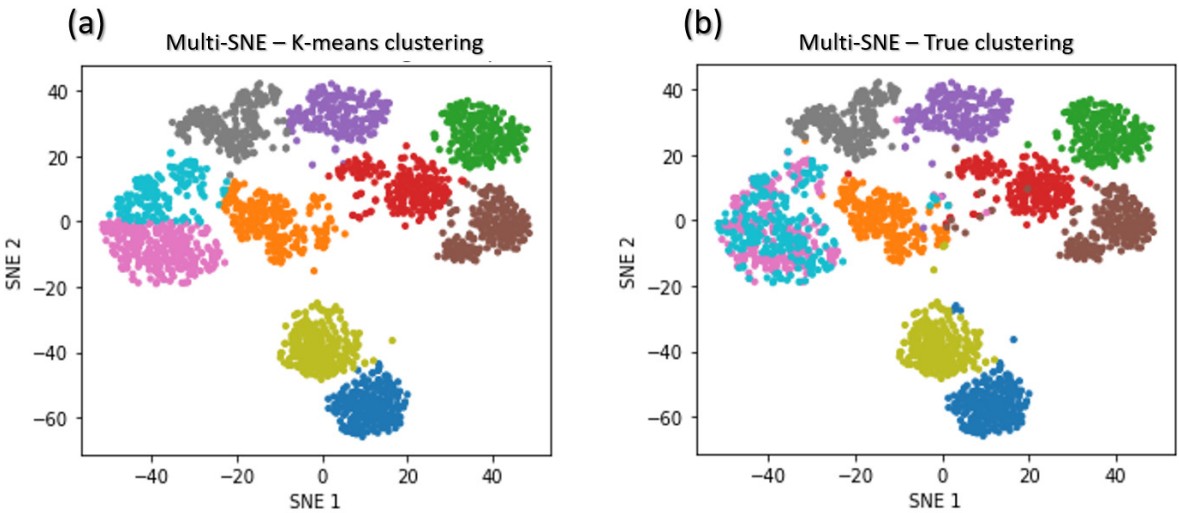

Figure 4: **Multi-SNE visualisations of handwritten digits**. Projections produced by multi-SNE with perplexity $Perp = 10$ . Colours present the clustering on the data points by **(a)** $K$-means, and **(b)** Ground truth.

embeddings can separate the data points to their respective clusters quantitatively with high accuracy via a clustering algorithm, then those clusters are expected to be qualitatively separated and visually shown in two

dimensions. For all examined data sets (synthetic and real), the number of clusters (ground truth) within the samples is known, which attracts the implementation of $K$-means over alternative clustering algorithms. The number of clusters was used as the input parameter, $K$, of the $K$-means algorithm and by computing the clustering measures we evaluated whether the correct sample allocations were made.

The proposed multi-SNE, multi-LLE, and multi-ISOMAP approaches were found to outperform their competitive multi-view extensions (m-SNE, m-LLE, m-ISOMAP) as well as their concatenated versions (**SNE**$_{concat}$, **LLE**$_{concat}$, **ISOMAP**$_{concat}$) (Tables 2 and 3). For the majority of the data sets the multi-SNE approach was found to overall outperform all other approaches.

Figure 4 shows a comparison between the true clusters of the handwritten digits data set and the clusters identified by $K$-means. The clusters reflecting the digits *6* and *9* are clustered together, but all remaining clusters are well separated and agree with the truth.

Multi-SNE applied on caltech7 produces a good visualisation, with clusters A and B being clearly separated from the rest (Figure 5b). Clusters C and G are also well-separated, but the remaining three clusters are bundled together. Applying $K$-means to that low-dimensional embedding does not capture the true structure of the data (Table 2). It provides a solution with all clusters being equally sized (Figure 5a) and thus its quantitative evaluation is misleading. Motivated by this result, we have further explored the performance of proposed approaches on a balanced version of the caltech7 data set (generated as described in Section 3.2).

Similarly to the visualisation of the original data set, the visualisation of the balanced caltech7 data set shows clusters A, B, C and G to be well-separated, while the remaining are still bundled together (Figures 5c and 5d).

Through the conducted work, it was shown that the multi-view approaches proposed in the manuscript generate low-dimensional embeddings that can be used as input features in a clustering algorithm (as for example the $K$-means algorithm) for identifying clusters that exist within the data set. We have illustrated that the proposed approaches outperform existing multi-view approaches and the visualisations produced by multi-SNE are very close to the ground truth of the data sets.

Alternative clustering algorithms, that do not require the number of clusters as input, can be considered as well. For example, Density-based spatial clustering of applications with noise (DBSCAN) measures the density around each data point and does not require the true number of clusters as input (Ester et al., 1996). In situations, where the true number of clusters is unknown, DBSCAN would be preferable over $K$-means. For completeness of our work, DBSCAN was applied on two of the real data sets explored, with similar results observed as the ones with $K$-means. The proposed multi-SNE approach was the best-performing method of partitioning the data samples. The analysis using DBSCAN can be found in Appendix D.8.

An important observation made was that caution needs to be taken when data sets with imbalanced clusters are analysed as the quantitative performance of the approaches on such data sets is not very robust.

### 4.3 Optimal Parameter Selection

SNE, LLE and ISOMAP depend on a parameter that requires tuning. Even though the parameter is defined differently in each algorithm, it is always related to the nearest number of global neighbours. As described earlier, the optimal parameter was found by comparing the performance of the methods on a range of parameter values, $S = \{2, 10, 20, 50, 80, 100, 200\}$. In this section, the synthetic data sets, NDS and MCS, were analysed, because both data sets separate the samples into known clusters by design and evaluation via clustering measures would be appropriate.

To find the optimal parameter value, the performance of the algorithms was evaluated by applying $K$-means on the low-dimensional embeddings and comparing the resulting clusterings against the truth. Once the optimal parameter was found, we confirmed that the clusters were visually separated by manually looking at the two-dimensional embeddings. Since the data in NDS and MCS are perfectly balanced and were generated for clustering, this approach can effectively evaluate the data visualisations.

On NDS, single-view SNE, LLE and ISOMAP algorithms produced a misclustering error of 0.3, meaning that a third of the samples was incorrectly clustered (Figure 6b). This observation shows that single-view

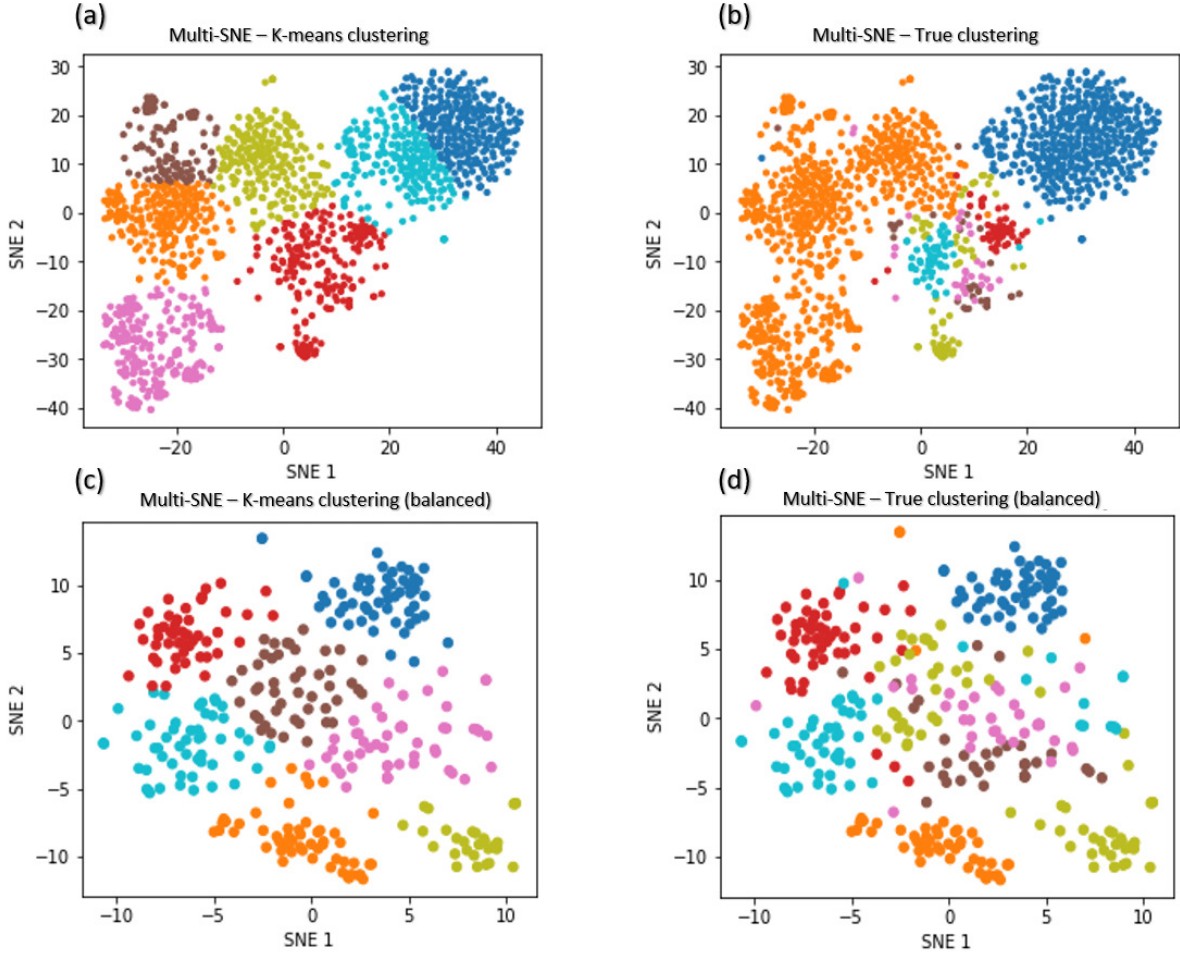

Figure 5: **Multi-SNE visualisations of caltech7 and its balanced subset.** Projections produced by multi-SNE with perplexity $Perp = 80$ and $Perp = 10$ for the original and balanced caltech7 data set, respectively. Colours present the clustering on the data points by **(a)**, **(c)** $K$-means, and **(b)**, **(d)** Ground truth. **(a) (b)** present the data points on the original caltech7 data set, while **(c)**, **(d)** are on its balanced subset.

methods capture the true local underlying structure of each synthetic data-view. The only exception for NDS is the fourth data-view, for which the error is closer to 0.6, *i.e.* randomly assigns the clusters (which follows the simulation design, as it was designed to be a random data-view). After concatenating the features of all data-views, the performance of single-view approaches remains poor (Figure 6a). The variance of the misclustering error on this solution is much greater, suggesting that single-view manifold learning algorithms on concatenated data are not robust and thus not reliable. Increasing the noise level (either by incorporating additional noisy data-views, or by increasing the dimensions of the noisy data-view) in this synthetic data set had little effect on the overall performance of the multi-view approaches (see Appendix D.4 for more information).

On both NDS and MCS, multi-LLE and multi-SNE were found to be sensitive to choice of their corresponding parameter value (Figures 6a and 7). While multi-LLE performed the best when the number of nearest neighbours was low, multi-SNE provided better results as perplexity was increasing. On the other hand, multi-ISOMAP had the highest NMI value when the parameter was high.

Overall, ISOMAP-based multi-view algorithms showed higher variability than the other multi-view methods, which makes them less favourable solutions. The performance of ISOMAP-based methods improved as the

| Data Set | Algorithm | Accuracy | NMI | RI | ARI |
|---|---|---|---|---|---|
| | SNE$_{concat}$ [Perp=10] | 0.717 (0.032) | 0.663 (0.013) | 0.838 (0.005) | 0.568 (0.026) |
| | m-SNE [Perp=10] | 0.776 (0.019) | 0.763 (0.009) | 0.938 (0.004) | 0.669 (0.019) |
| | **multi-SNE** [Perp=10] | 0.882 (0.008) | 0.900 (0.005) | 0.969 (0.002) | 0.823 (0.008) |
| | LLE$_{concat}$ [NN=10] | 0.562 | 0.560 | 0.871 | 0.441 |
| Handwritten Digits | m-LLE [NN=10] | 0.632 | 0.612 | 0.896 | 0.503 |
| | multi-LLE [NN=5] | 0.614 | 0.645 | 0.897 | 0.524 |
| | ISOMAP$_{concat}$ [NN=20] | 0.634 | 0.619 | 0.905 | 0.502 |
| | m-ISOMAP [NN=20] | 0.636 | 0.628 | 0.898 | 0.477 |
| | multi-ISOMAP [NN=5] | 0.658 | 0.631 | 0.909 | 0.518 |
| | SNE$_{concat}$ [Perp=50] | 0.470 (0.065) | 0.323 (0.011) | 0.698 (0.013) | 0.290 (0.034) |
| | **m-SNE** [Perp=10] | 0.542 (0.013) | 0.504 (0.029) | 0.757 (0.010) | 0.426 (0.023) |
| | multi-SNE [Perp=80] | 0.506 (0.035) | 0.506 (0.006) | 0.754 (0.009) | 0.428 (0.022) |
| | LLE$_{concat}$ [NN=100] | 0.425 | 0.372 | 0.707 | 0.305 |
| Caltech7 | m-LLE [NN=5] | 0.561 | 0.348 | 0.718 | 0.356 |
| | multi-LLE [NN=80] | 0.638 | 0.490 | 0.732 | 0.419 |
| | ISOMAP$_{concat}$ [NN=20] | 0.408 | 0.167 | 0.634 | 0.151 |
| | m-ISOMAP [NN=5] | 0.416 | 0.306 | 0.686 | 0.261 |
| | multi-ISOMAP [NN=10] | 0.519 | 0.355 | 0.728 | 0.369 |
| | SNE$_{concat}$ [Perp=80] | 0.492 (0.024) | 0.326 (0.018) | 0.687 (0.023) | 0.325 (0.015) |
| | m-SNE [Perp=10] | 0.581 (0.011) | 0.444 (0.013) | 0.838 (0.022) | 0.342 (0.016) |
| | **multi-SNE** [Perp=20] | 0.749 (0.008) | 0.686 (0.016) | 0.905 (0.004) | 0.619 (0.009) |
| | LLE$_{concat}$ [NN=20] | 0.567 | 0.348 | 0.725 | 0.380 |
| Caltech7 (balanced) | m-LLE [NN=10] | 0.403 | 0.169 | 0.617 | 0.139 |
| | multi-LLE [NN=5] | 0.622 | 0.454 | 0.710 | 0.391 |
| | ISOMAP$_{concat}$ [NN=5] | 0.434 | 0.320 | 0.791 | 0.208 |
| | m-ISOMAP [NN=5] | 0.455 | 0.299 | 0.797 | 0.224 |
| | multi-ISOMAP [NN=5] | 0.548 | 0.368 | 0.810 | 0.267 |
| | SNE$_{concat}$ [Perp=10] | 0.625 (0.143) | 0.363 (0.184) | 0.301 (0.113) | 0.687 (0.169) |
| | m-SNE [Perp=10] | 0.923 (0.010) | 0.839 (0.018) | 0.876 (0.011) | 0.922 (0.014) |
| | **multi-SNE** [Perp=20] | 0.964 (0.007) | 0.866 (0.023) | 0.902 (0.005) | 0.956 (0.008) |
| | LLE$_{concat}$ [NN=10] | 0.502 | 0.122 | 0.091 | 0.576 |
| Cancer types | m-LLE [NN=20] | 0.637 | 0.253 | 0.235 | 0.647 |
| | multi-LLE [NN=10] | 0.850 | 0.567 | 0.614 | 0.826 |
| | ISOMAP$_{concat}$ [NN=5] | 0.384 | 0.015 | 0.009 | 0.556 |
| | m-ISOMAP [NN=10] | 0.390 | 0.020 | 0.013 | 0.558 |
| | multi-ISOMAP [NN=50] | 0.514 | 0.116 | 0.093 | 0.592 |

Table 2: **Clustering performance.** For each data set, red highlights the method with the best performance on each measure between each group of algorithms (SNE, LLE or ISOMAP based). The overall superior method for each data set is depicted with **bold**. The parameters $Perp$ and $NN$ refer to the selected perplexity and number of nearest neighbours, respectively. They were optimised for the corresponding methods. Due to the non-convexity of SNE-based approaches, the mean (and standard deviation) of 100 separate runs on the same data is reported.

parameter value increased (Figure 7). However, they were outperformed by multi-LLE and multi-SNE for both synthetic data sets.

Out of the three manifold learning foundations, LLE-based approaches mostly depend on their parameter value to produce the optimal outcome. Specifically, their performance dropped when the parameter value lay between 20 and 100 (Figure 6a). When the number of nearest neighbours was set to be greater than 100 their performance started to improve. Out of all LLE-based algorithms, the highest NMI and lowest misclustering error was obtained by multi-LLE (Figures 6 and 7). Our observations on the tuning parameters of LLE-based approaches are in agreement with earlier studies (Karbauskaitė et al., 2007; Valencia-Aguirre et al., 2009). Both Karbauskaitė et al. (2007) and Valencia-Aguirre et al. (2009) found that LLE performs best with low nearest number of neighbours and their conclusions reflect the performance of multi-LLE; best performed on low values of the tuning parameter. Even though m-SNE performed better than single-view methods in terms of both clustering and error variability, multi-SNE produced the best results (Figures 6 and 7). In particular, multi-SNE outperformed all algorithms presented in this paper on both NDS and

| Data Set | Algorithm | Accuracy | NMI | RI | ARI |
|---|---|---|---|---|---|
| NDS | SNE$_{concat}$ [Perp=80] | 0.747 (0.210) | 0.628 (0.309) | 0.817 (0.324) | 0.598 (0.145) |
| | m-SNE [Perp=50] | 0.650 (0.014) | 0.748 (0.069) | 0.766 (0.022 | 0.629 (0.020) |
| | **multi-SNE** [Perp=80] | **0.989** (0.006) | **0.951** (0.029) | **0.969** (0.019) | **0.987** (0.009) |
| | LLE$_{concat}$ [NN=5] | 0.606 (0.276) | 0.477 (0.357) | 0.684 (0.359) | 0.446 (0.218) |
| | m-LLE [NN=20] | 0.685 (0.115) | 0.555 (0.134) | 0.768 (0.151) | 0.528 (0.072)) |
| | multi-LLE [NN=20] | 0.937 (0.044) | 0.768 (0.042) | 0.922 (0.028) | 0.823 (0.047) |
| | ISOMAP$_{concat}$ [NN=100] | 0.649 (0.212) | 0.528 (0.265) | 0.750 (0.286) | 0.475 (0.133) |
| | m-ISOMAP [NN=5] | 0.610 (0.234) | 0.453 (0.221) | 0.760 (0.280) | 0.386 (0.138) |
| | multi-ISOMAP [NN=300] | 0.778 (0.112) | 0.788 (0.234) | 0.867 (0.194) | 0.730 (0.094) |
| MCS | SNE$_{concat}$ [Perp=200] | 0.421 (0.200) | 0.215 (0.185) | 0.711 (0.219) | 0.173 (0.089) |
| | m-SNE [Perp=2] | 0.641 (0.069) | 0.670 (0.034) | 0.854 (0.080) | 0.575 (0.055) |
| | **multi-SNE** [Perp=50] | **0.919** (0.046) | **0.862** (0.037) | **0.942** (0.052) | **0.819** (0.018) |
| | LLE$_{concat}$ [NN=50] | 0.569 (0.117) | 0.533 (0.117) | 0.796 (0.123) | 0.432 (0.051) |
| | m-LLE [NN=20] | 0.540 (0.079) | 0.627 (0.051) | 0.819 (0.077) | 0.487 (0.026) |
| | multi-LLE [NN=20] | 0.798 (0.059) | 0.647 (0.048) | 0.872 (0.064) | 0.607 (0.022) |
| | ISOMAP$_{concat}$ [NN=150] | 0.628 (0.149) | 0.636 (0.139) | 0.834 (0.167) | 0.526 (0.071) |
| | m-ISOMAP [NN=5] | 0.686 (0.113) | 0.660 (0.106) | 0.841 (0.119) | 0.565 (0.051) |
| | multi-ISOMAP [NN=300] | 0.717 (0.094) | 0.630 (0.101) | 0.852 (0.118) | 0.570 (0.044) |

Table 3: **Clustering performance.** For each data set, red highlights the method with the best performance on each measure between each group of algorithms (SNE, LLE or ISOMAP based). The overall superior method for each data set is depicted with **bold**. The parameters $Perp$ and $NN$ refer to the selected perplexity and number of nearest neighbours, respectively. They were optimised for the corresponding methods.

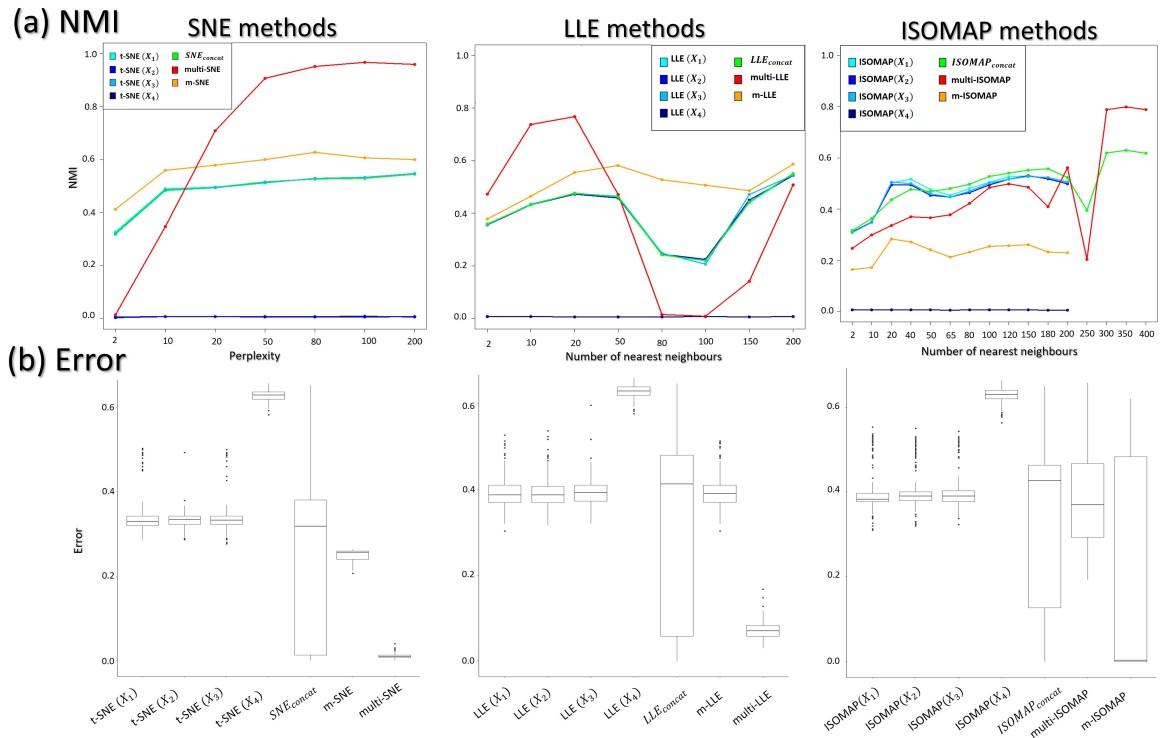

Figure 6: **NDS evaluation measures.** **(a)** NMI values along different parameter values on all manifold learning algorithms and **(b)** Misclustering error on the optimal parameter values.

MCS. Even though it performed poorly for low perplexity values, its performance improved for $Perp \geq 20$. Multi-SNE was the algorithm with the lowest error variance, making it a robust and preferable solution.

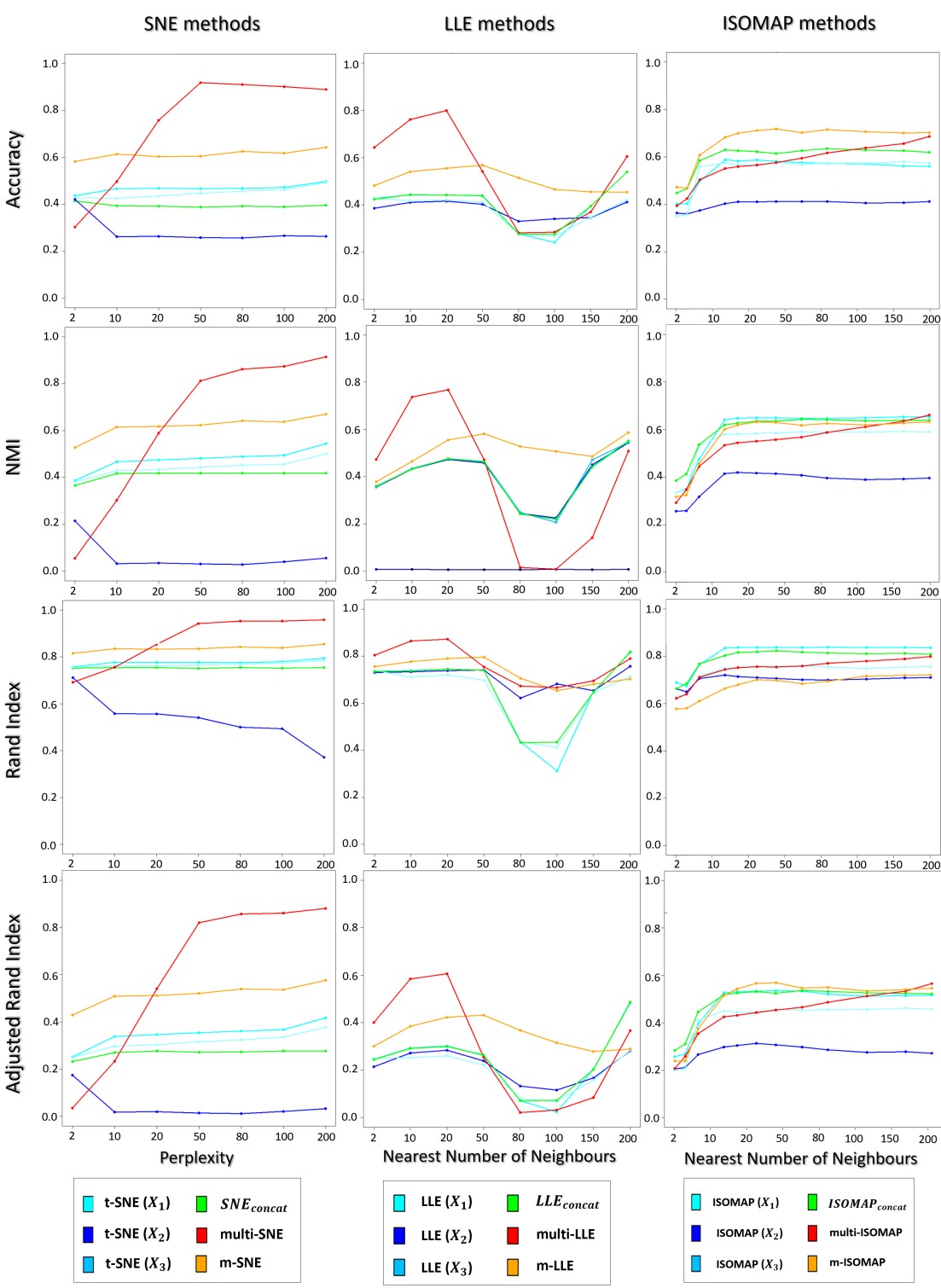

Figure 7: **MCS evaluation measures.** The clustering evaluation measures are plotted against different parameter values on the all SNE, LLE and ISOMAP based algorithms.

The four implemented measures (Accuracy, NMI, RI and ARI) use the true clusters of the samples to evaluate the clustering performance. In situations where cluster allocation is unknown, alternative clustering evaluation measures can be used, such as the Silhouette score (Rousseeuw, 1987). The Silhouette score in contrast to the other measures does not require as input the cluster allocation and is a widely used approach for identifying the best number of clusters and clustering allocation in an unsupervised setting.

Evaluating the clustering performance of the methods via the Silhouette score agrees with the other four evaluation measures, with multi-SNE producing the highest value out of all multi-view manifold learning solutions. The Silhouette score of all methods applied on the MCS data set can be found in Appendix D.7.

The same process of parameter tuning was implemented for the real data sets and their performance is presented in the Appendix D.3. In contrast to the synthetic data, multi-SNE on cancer types data performed the best at low perplexity values. For the remaining data sets, its performance was stable for all parameter values. With the exception of cancer types data, the performance of LLE-based solutions follows their behaviour on synthetic data.

### 4.4 Optimal Number of Data-views

It is common to think that more information would lead to better results, and in theory that should be the case. However, in practice that is not always true (Kumar et al., 2011). Using the cancer types data set, we explored whether the visualisations and clusterings are improved if all or a subset of the data-views are used. With three available data-views, we implemented a multi-view visualisation on three combinations of two data-views and a single combination of three data-views.

The genomics data-view provides a reasonably good separation of the three cancer types, whereas miRNA data-view fails in this task, as it provides a visualisation that reflects random noise (first column of plots in Figure 8). This observation is validated quantitatively by evaluating the produced t-SNE embeddings (Table 8 in Appendix D.5). Concatenating features from the different data-views before implementing t-SNE does not improve the final outcome of the algorithm, regardless of the data-view combination.

Overall, multi-view manifold learning algorithms have improved the data visualisation to a great extent. When all three data-views are considered, both multi-SNE and m-SNE provide a good separation of the clusters (Figure 8). However, the true cancer types can be identified perfectly when the miRNA data-view is discarded. In other words, the optimal solution in this data set is obtained when only genomics and epigenomics data-views are used. That is because miRNA data-view contains little information about the cancer types and adds random noise, which makes the task of separating the data points more difficult.

This observation was also noted between the visualisations of MMDS and NDS (Figure 9). The only difference between the two synthetic data sets is the additional noisy data-view in NDS. Even though NDS separates the samples to their corresponding clusters, the separation is not as clear as it is in the projection of MMDS via multi-SNE. In agreement with the exploration of the cancer types data set, it is favourable to discard any noisy data-views in the implementation of multi-view manifold learning approaches.

It is not always a good idea to include all available data-views in multi-view manifold learning algorithms; some data-views may provide noise which would result in a worse visualisation than discarding those data-views entirely. The noise of a data-view with unknown labels may be viewed in a single-view t-SNE plot (all data-points in a single cluster), or identified, if possible, via quantification measures such as signal-to-noise ratio.

### 4.5 Multi-SNE variations

This section presents two alternative variations of multi-SNE, including automatic weight adjustments and multi-CCA as a pre-training step for reducing the dimensions of the input data-views.

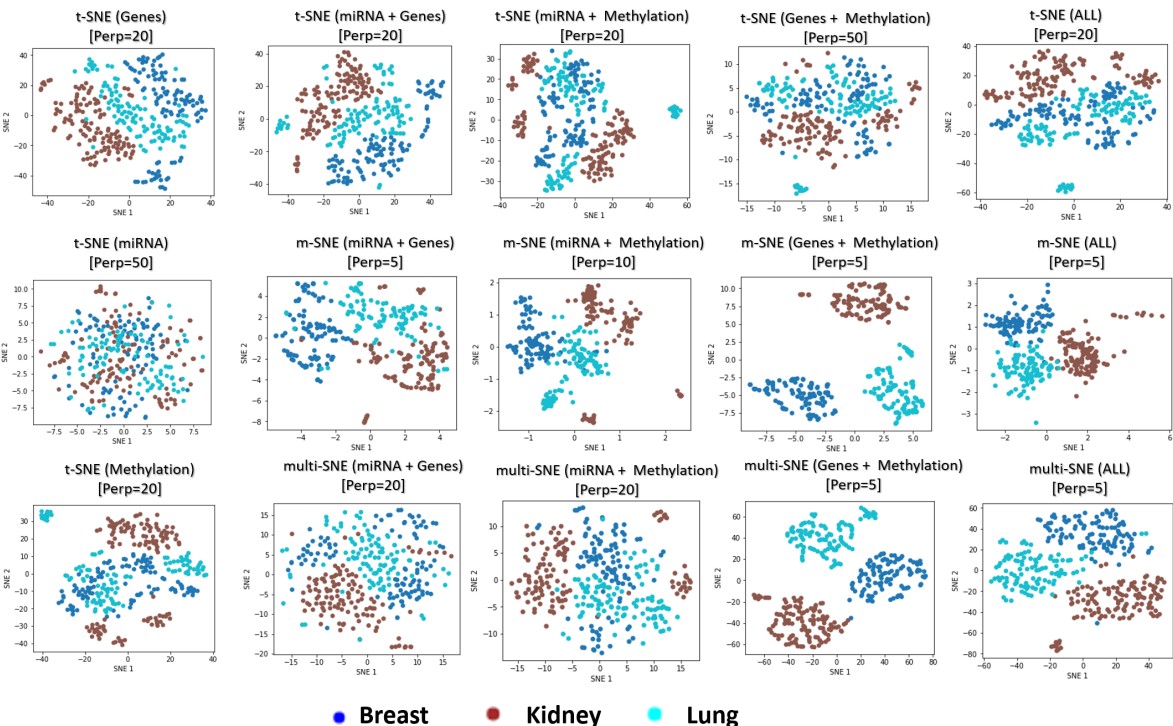

Figure 8: **Visualisations of cancer types.** Projections produced by all SNE-based manifold learning algorithms on all possible combinations between the three data-views in the cancer types data set.The parameter *Perp* refers to the selected perplexity, which was optimised for the corresponding methods.

### 4.5.1 Automated weight adjustments

A simple weight-updating approach is proposed based on the KL-divergence measure from each data-view. This simple weight-updating approach guarantees that more weight is given to the data-views producing lower KL-divergence measures and that no data-view is being completely discarded from the algorithm.

Recall that $KL(P||Q) \in [0, \infty)$, with $KL(P||Q) = 0$, if the two distributions, $P$ and $Q$, are perfectly matched. Let $\mathbf{k} = (k^{(1)}, \cdots, k^{(M)})$ be a vector, where $k^{(m)} = KL(P^{(m)}||Q), \forall m = \{1, \cdots, M\}$ and initialise the weight vector $\mathbf{w} = (w^{(1)}, \cdots, w^{(M)})$ by $w^{(m)} = \frac{1}{M}, \forall m$. To adjust the weights of each data-view, the following steps are performed at each iteration:

1. Normalise KL-divergence by $k^{(m)} = \frac{k^{(m)}}{\sum_i^M k^{(i)}}$. This step ensures that $k^{(m)} \in [0, 1], \forall m$ and that $\sum_m k^{(m)} = 1$.

2. Measure the weights for each data-view by $w^{(m)} = 1 - k^{(m)}$. This step ensures that the data-view with the lowest KL-divergence value receives the highest weight.

Based on the analysis in Section 4.4, we know that cancer types and NDS data sets contain noisy data-views and thus multi-SNE performs better when they are entirely discarded. Here, we assume that this information is unknown and the proposed weight-updating approach is implemented on those two data sets to test if the weights are being adjusted correctly according to the noise level of each data-view.

The proposed weight-adjustment process, which looks at the produced KL-divergence between each data-view and the low-dimensional embeddings, distinguishes which data-views contain the most noise and the weight values are updated accordingly (Figure 10b). In cancer types, transcriptomics (miRNA) receives the lowest weight, while genomics (Genes) was given the highest value. This weight adjustment comes in

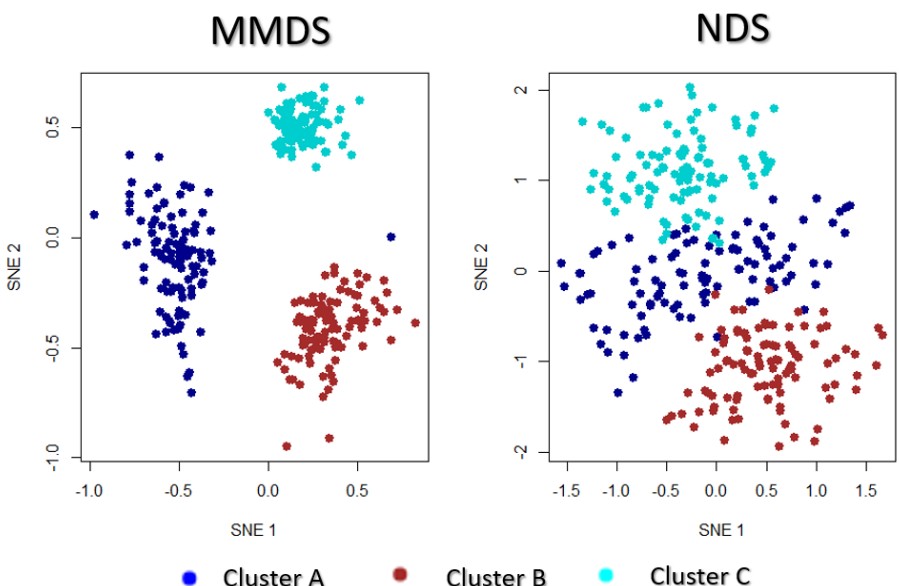

Figure 9: **Multi-SNE visualisations of MMDS and NDS.** Projections produced by multi-SNE with perplexity $Perp = 100$ for both MMDS and NDS.

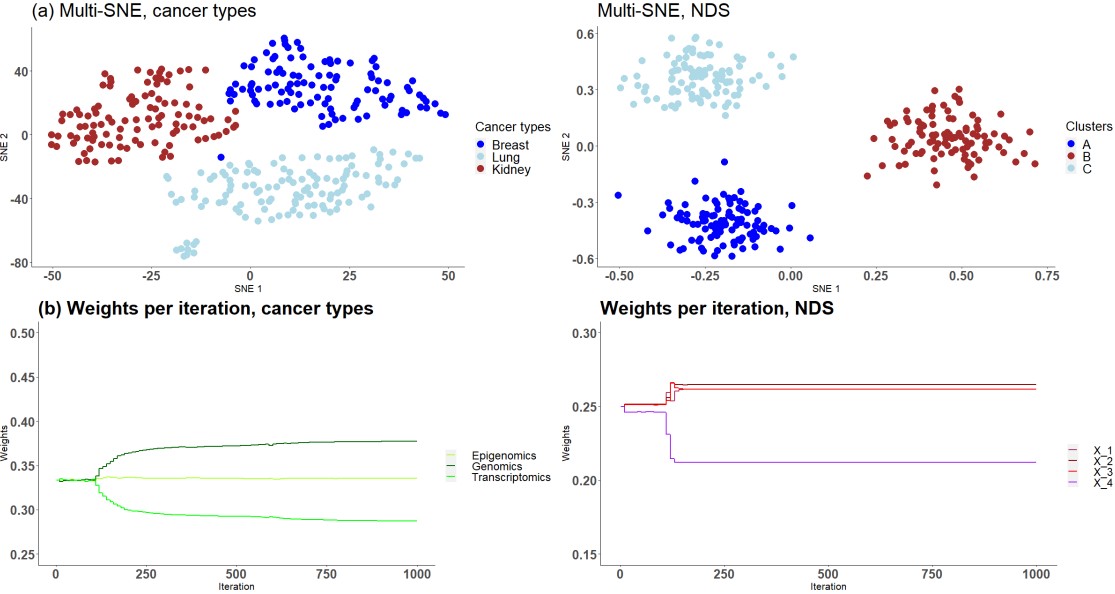

Figure 10: **Cancer types and NDS with automated weight adjustments.** The first row presents the produced visualisations of multi-SNE with the automated weight adjustment procedure implemented. The second row of figures presents the weights assigned to each data-view at each step of the iteration. For both data the iterations ran for a maximum of 1,000 steps.

agreement with the qualitative (t-SNE plots) and quantitative (clustering) evaluations performed in Section 4.4. In NDS, $X^{(4)}$ which represents the noisy data-view received the lowest weight, and the other data-views had around the same weight value, as they all impact the final outcome equally.

The proposed weight-adjustment process updates the weights at each iteration. For the first 100 iterations, the weights are not changing, as the algorithm adjusts to the produced low-dimensional embeddings (Figure

10b). In NDS, the weights converge after 250 iterations, while in cancer types, they are still being updated even after 1000 iterations. The changes recorded are small and the weights can be said to have stabilised.

The low-dimensional embeddings produced in NDS with weight adjustments separate clearly the three clusters, an observation missed without the implementation of the weight-updating approach (Figure 10a); it resembles the MMDS (*i.e.* without noisy data-view) multi-SNE plot (Figure 9). The automatic weight-adjustment process identifies the informative data-views, by allocating them a higher weight value than to the noisy data-views. This observation was found to be true even when a dataset contains more noise than informative data-views (see Appendix D.4 for further details).

The produced embeddings in cancer types do not separate the three clusters as clearly as multi-SNE without the noisy data-view, but it projects a more clear separation than multi-SNE on the complete data set without weight adjustments.

The weights produced by this weight adjustment approach can indicate the importance of each data-view in the final lower-dimensional embedding. For example, data-views with very low weights may be assumed futile and a better visualisation may be produced if those data-views are discarded. The actual weights assigned to each data-view do not have any further meaning.

### 4.5.2 Multi-CCA as pre-training

As mentioned earlier, van der Maaten & Hinton (2008) proposed the implementation of PCA as a pre-training step for t-SNE to reduce the computational costs, provided that the fraction of variance explained by the principal components is high. In this paper, pre-training via PCA was implemented in all variations of SNE. Alternative linear dimensionality reduction methods may be considered, especially for multi-view data. In addition to reducing the dimensions of the original data, such methods can capture information between the data-views. For example, Canonical Correlation Analysis (CCA) captures relationships between the features of two data-views by producing two latent low-dimensional embeddings (canonical vectors) that are maximally correlated between them (Hotelling, 1936; Rodosthenous et al., 2020). Rodosthenous et al. (2020) demonstrated that multi-CCA, an extension of CCA that analyses multiple (more than two) data-views, would be preferable as it reduces over-fitting.

This section demonstrates the application of multi-CCA as pre-training in replacement of PCA. This alteration of the multi-SNE algorithm was implemented on the handwritten digits data set. Multi-CCA was applied on all data-views, with 6 canonical vectors produced for each data-view (in this particular data set $\min(p_1, p_2, p_3, p_4, p_5, p_6) = 6$). The variation of multi-CCA proposed by Witten & Tibshirani (2009) was used for the production of the canonical vectors, as it is computationally cheaper compared to others (Rodosthenous et al., 2020). By using these vectors as input features, multi-SNE produced a qualitatively better visualisation than using the principal components as input features (Figure 11). By using an integrative algorithm as pre-training, all 10 clusters are clearly separated, including 6 and 9. Quantitatively, clustering via $K$-Means was evaluated with ACC = 0.914, NMI = 0.838, RI = 0.968, ARI = 0.824. This evaluation suggests that quantitatively, it performed better than the 10-dimensional embeddings produced multi-SNE with PCA as pre-training.

### 4.5.3 Comparison of multi-SNE variations

Section 2.2 introduced multi-SNE, a multi-view extension of t-SNE. In Sections 4.5.1 and 4.5.2, two variations of multi-SNE are presented. The former implements a weight-adjustment process which at each iteration updates the weights allocated for each data-view, and the latter uses multi-CCA instead of PCA as a pre-training step. In this section, multi-SNE and its two variations are compared to assess whether the variations introduced to the algorithm perform better than the initial proposal.

The implementation of the weight-adjustment process improved the performance of multi-SNE on all real data sets analysed (Table 4). The influence of multi-CCA as a pre-training step produced inconsistent results; in some data sets this step boosted the clustering performance of multi-SNE (*e.g.* handwritten digits), while for the other data sets, it did not (*e.g.* cancer types). From this analysis, we conclude that adjusting the

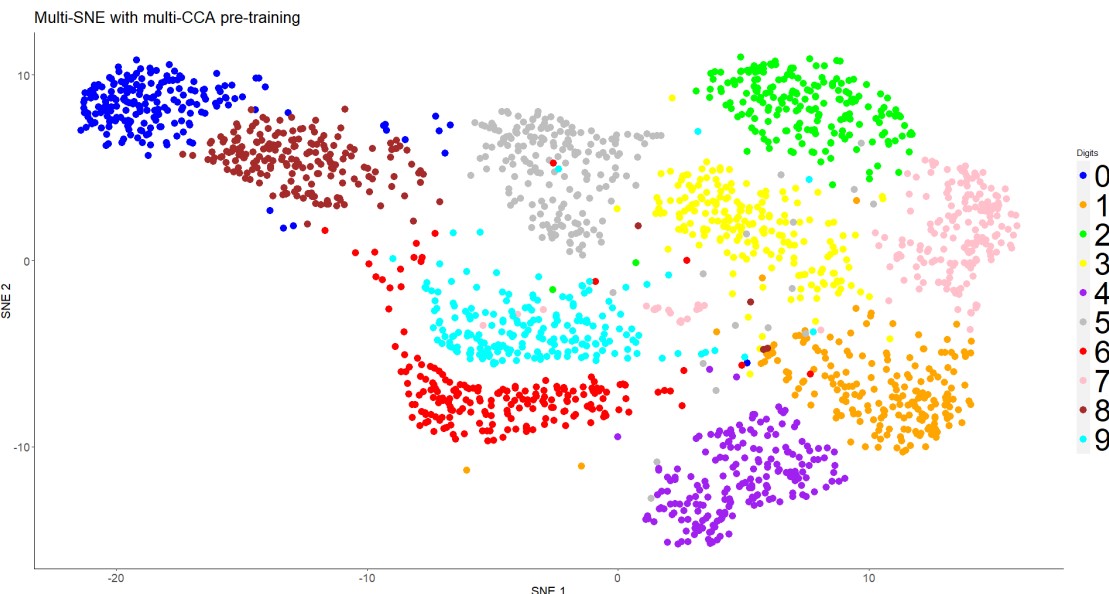

Figure 11: **Multi-SNE on handwritten digits, with multi-CCA as pre-training.** Projections of handwritten digits data sets, produced by multi-SNE. Multi-CCA was implemented on all data-views with their respective canonical vectors acting as input features for multi-SNE.

| Variation | Handwritten digits | Cancer types | Caltech7 original | Caltech7 balanced | NDS | MCS |
|---|---|---|---|---|---|---|
| Multi-SNE without weight-adjustment | 0.822 | 0.964 | 0.506 | 0.733 | 0.989 (0.006) | 0.919 (0.046) |
| Multi-SNE with weight-adjustment | 0.883 | **0.994** | **0.543** | 0.742 | **0.999 (0.002)** | 0.922 (0.019) |
| Multi-CCA multi-SNE without weight-adjustment | 0.901 | 0.526 | 0.453 | 0.713 | 0.996(0.002) | **0.993 (0.005)** |
| Multi-CCA multi-SNE with weight-adjustment | **0.914** | 0.562 | 0.463 | **0.754** | 0.996 (0.002) | **0.993 (0.005)** |

Table 4: **Clustering performance of multi-SNE variations.** For each data set, **bold** highlights the multi-SNE variation with the best performance - highest accuracy (ACC). Perplexity was optimised for all variations. The mean performance (and its standard deviation) is depicted for the synthetic data sets NDS and MCS.)

weights of each data-view always improves the performance of multi-SNE. On the other hand, the choice of pre-training, either via PCA or multi-CCA, is not clear, and it depends on the data at hand.

## 5  Discussion

In this manuscript, we propose extensions of the well-known manifold learning approaches t-SNE, LLE, and ISOMAP for the visualisation of multi-view data sets. These three approaches are widely used for the visualisation of high-dimensional and complex data sets on performing non-linear dimensionality reduction. The increasing number of multiple data sets produced for the same samples in different fields, emphasises the need for approaches that produce expressive presentations of the data. We have illustrated that visualising each data set separately from the rest is not ideal as it does not reveal the underlying patterns within the samples. In contrast, the proposed multi-view approaches can produce a single visualisation of the samples by integrating all available information from the multiple data-views. `Python` and `R` (only for multi-SNE) code of the proposed solutions can be found in the links provided in Appendix E.

Multi-view visualisation has been explored in the literature with a number of approaches proposed in recent years. In this work, we propose multi-view visualisation approaches that extend the well-known manifold approaches: t-SNE, LLE, and ISOMAP. Through a comparative study of real and synthetic data, we have illustrated that the proposed approach, multi-SNE, provides a better and more robust solution compared to the other tested approaches proposed in the manuscript (multi-LLE and multi-ISOMAP) and the approaches proposed in the literature including m-LLE, m-SNE, MV-tSNE2, j-SNE, j-UMAP (additional results in Appendices D.1, D.2). Although multi-SNE was computationally the most expensive multi-view manifold learning algorithm (Table 9 in Appendix F), it was found to be the solution with the superior performance, both qualitatively and quantitatively.

We have utilised the low-dimensional embeddings of the proposed algorithms as features in the $K$-means clustering algorithm, which we have used (1) to quantify the visualisations produced, and (2) to select the optimal tuning parameters for the manifold learning approaches. By investigating synthetic and real multi-view data sets, each with different data characteristics, we concluded that multi-SNE provides a more accurate and robust solution than any other single-view and multi-view manifold learning algorithms we have considered. Specifically, multi-SNE was able to produce the best data visualisations of all data sets analysed in this paper. Multi-LLE provides the second-best solution, while multi-view ISOMAP algorithms have not produced competitive visualisations. By exploring several data sets, we concluded that multi-view manifold learning approaches can be effectively applied to heterogeneous and high-dimensional data (*i.e.* $p \gg n$).

Through the conducted experiments, we have illustrated the effect of the parameters on the performance of the methods. We have shown that SNE-based methods perform the best when perplexity is in the range $[20, 100]$, LLE-based algorithms should take a small number of nearest neighbours, in the range $[10, 50]$, while the parameter of ISOMAP-based should be in the range $[100, N]$, where $N$ is the number of samples.

We believe that the best approach to selecting the tuning parameters of the methods is to explore a wide range of different parameter values and assess the performance of the methods both qualitatively and quantitatively. If the produced visualisations vary a lot between a range of parameter values, then the data might be too noisy, and the projections misleading. In this case, it might be beneficial to look at the weights obtained for each data-view and explore removing the noisiest data-views (depending on the number of data-views used and/or existing knowledge of noise in the data). Otherwise (if the produced visualisations vary slightly between various parameter values), the parameter value with the best qualitative and quantitative performance can be selected. Since t-SNE (and its extensions) are robust to perplexity (van der Maaten & Hinton, 2008), a strict optimal parameter value would not be necessary to produce meaningful visualisations and clusterings, *i.e.* identical performance qualitatively and quantitatively can be observed for a range of values.

Cao & Wang (2017) proposed an automatic approach for selecting the perplexity parameter of t-SNE. According to the authors the trade between the final KL divergence and perplexity value can lead to good embeddings, and they proposed the following criterion:

$$S(Perp) = 2KL(P||Q) + \log(n)\frac{Perp}{n} \tag{10}$$

This solution can be extended to automatically select the multi-SNE perplexity, by modifying the criterion to:

$$S(Perp) = 2\sum_m KL(P^{(m)}||Q) + \log(n)\frac{Perp}{n} \tag{11}$$

Our conclusions about the superiority of multi-SNE have been further supported by implementing the Silhouette score as an alternative approach for evaluating the clustering and tuning the parameters of the methods. In contrast to the measures used throughout the paper, the Silhouette score does not take into account the number of clusters that exist in the data set, illustrating the applicability of multi-SNE approach in unsupervised learning problems where the underlying clusters of the samples are not known (Appendix

**Multi-view Clustering on handwritten digits data set**

| | Kumar et al. (2011) | Liu et al. (2013) | Sun et al. (2015) | Ou et al. (2016) | Ou et al. (2018) | multi-SNE with PCA/multi-CCA | | | |
| --- | --- | --- | --- | --- | --- | --- | --- | --- | --- |
| | | | | | | 2D | 3D | 5D | 10D |
| NMI | 0.768 | 0.804 | 0.876 | 0.785 | 0.804 | 0.863/0.838 | 0.894/0.841 | **0.897**/0.848 | **0.899**/0.850 |
| ACC | – | 0.881 | – | 0.876 | 0.880 | 0.822/0.914 | 0.848/0.915 | 0.854/**0.922** | 0.849/**0.924** |

Table 5: **Multi-view clustering performance on handwritten digits.** The NMI and accuracy (ACC) values of multi-view clustering approaches, as they were presented by the authors in their corresponding papers, are depicted along the clustering performance of multi-SNE on a range of dimensions for the embeddings ($d = 2, 3, 5, 10$). The performance of multi-SNE with PCA and multi-CCA are depicted in the table, with weight adjustments in both variations.

D.7). Similarly, we have illustrated that alternative clustering algorithms can be implemented for clustering the samples. By inputting the produced multi-SNE embeddings in the DBSCAN algorithm we further illustrated how the clusters of the samples can be identified (Appendix D.8).

Multi-view clustering is a topic that has gathered a lot of interest in recent years with a number of approaches published in the literature. Such approaches include the ones proposed by Kumar et al. (2011), Liu et al. (2013), Sun et al. (2015), Ou et al. (2016) and Ou et al. (2018). The handwritten data set presented in the manuscript has been analysed by the aforementioned studies for multi-view clustering. Table 5 shows the NMI and accuracy values of the clusterings performed by the multi-view clustering algorithms (these values are as given in the corresponding articles). In addition, the NMI and accuracy values of the $K$-means clustering applied on the multi-SNE low-dimensional embeddings (from 2 to 10 dimensions) are presented in the table. On handwritten digits, the multi-SNE variation with multi-CCA as pre-training and weight adjustments had the best performance (Table 4). This variation of multi-SNE with $K$-means was compared against the multi-view clustering algorithms and it was found to be the most accurate, while pre-training with PCA produced the highest NMI (Table 5). By applying $K$-means to the low-dimensional embeddings of multi-SNE can successfully cluster the observations of the data (see Appendix D.6 for a 3-dimensional visualisation via multi-SNE).

An important area of active current research, where manifold learning approaches, such as t-SNE, as visualisation tools are commonly used is single-cell sequencing (scRNA-seq) and genomics. Last few years, have seen fast developments of multi-omics single-cell methods, where for example for the same cells multiple omics measurements are being obtained such as transcripts by scRNA-seq and chromatin accessibility by a method known as scATAC-seq (Stuart et al., 2019). As recently discussed the integration of this kind of multi-view single-cell data poses unique and novel statistical challenges (Argelaguet et al., 2021). We, therefore, believe our proposed multi-view methods will be very useful in producing an integrated visualisation of cellular heterogeneity and cell types studied by multi-omics single-cell methods in different tissues, in health and disease.

To illustrate the capability of multi-SNE for multi-omics single-cell data, we applied multi-SNE on a representative data set of scRNA-seq and ATAC-seq for human peripheral blood mononuclear cells (PBMC) [5] (Figure 7). Multi-SNE produced more intelligible projections of the cells compared to m-SNE and achieved higher evaluation scores (Appendix C). To test the quality of the obtained multi-view visualisation, we compared its performance against the multi-view clustering approach proposed by Liu et al. (2013) on this single-cell data. A balanced subset of this data set was used, which consists of two data-views on 9105 cells (*scRNA-seq* and *ATAC-seq* with 36000 and 108000 features, respectively). A detailed description of this data set, the pre-processing steps performed, and the projections of t-SNE and multi-SNE on the original data are provided in the Appendix C (Figure 13). We found Multi-SNE to have the highest accuracy (and a close NMI to the approach by Liu et al. (2013)) as seen in Figure 7. Qualitatively, the projections by t-SNE on scRNA-seq and multi-SNE are similar, but multi-SNE separates the clusters better (especially between *CD4* and *CD8* cell types (Figure 12, Figure 13 in Appendix C). While it is known that ATAC-seq data is noisier and has less information by itself, we see that integration of the data-views results in better overall separation of the different cell types in this data set. These results indicate the promise of multi-SNE as a unified multi-view and clustering approach for multi-omics single-cell data.

---

[5] `https://support.10xgenomics.com/single-cell-multiome-atac-gex/datasets/1.0.0/pbmc_granulocyte_sorted_10k`

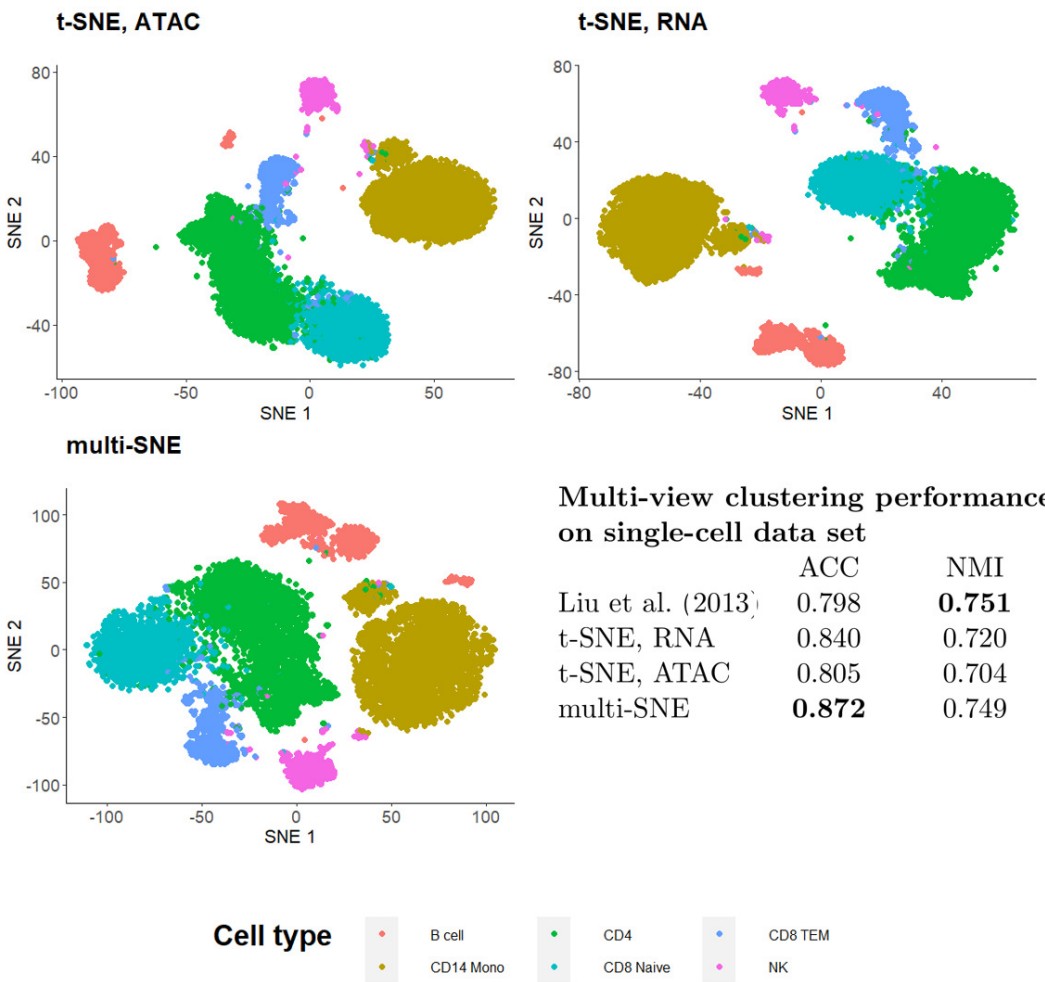

Figure 12: **Visualisations and clustering performance on single-cell multi-omics data.** Projections produced by t-SNE on RNA, ATAC and multi-SNE on both data-views with perplexity $Perp = 80$ for the two t-SNE projections and $Perp = 20$ for multi-SNE. The clustering performance of the data by Liu et al. (2013), t-SNE and multi-SNE are presented.

The increasing number of multi-view, high-dimensional and heterogeneous data requires novel visualisation techniques that integrate this data into expressive and revealing representations. In this manuscript, new multi-view manifold learning approaches are presented and their performance across real and synthetic data sets with different characteristics was explored. The multi-SNE approach is proposed to provide a unified solution for robust visualisation and subsequent clustering of multi-view data.

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

# A  Algorithms

**Data:** $M$ data sets, $X^{(m)} \in \mathbb{R}^{N \times p_m}, \quad \forall m \in \{1, \cdots, M\}$
**Parameters:** $Perp$ [Perplexity]; $T$ [Number of iterations]; $\eta$ [Learning rate]; $\alpha(t)$ [Momentum]
**Result:** Induced embedding, $Y \in \mathbb{R}^{n \times d}$. Often, $d = 2$
**begin**
    Optional step of implementing PCA, or multi-CCA on $X^{(m)} \quad \forall m \in \{1, \cdots, M\}$
    Compute pairwise affinities $p_{i|j}^m$ with perplexity $Perp, \quad \forall m \in \{1, \cdots, M\}$
    Set $p_{ij}^m = \frac{p_{i|j}^m + p_{j|i}^m}{2n}, \quad \forall m \in \{1, \cdots, M\}$
    Initialise solution $Y^{(0)} \sim \mathcal{N}(0, 0.1)$
    **for** *t=1 **to** T* **do**
        Compute induced affinities $q_{i|j}$ and set sum of gradients, $G = 0$
        **for** *m=1 **to** M* **do**
            Compute gradient $\frac{\delta C_m}{\delta Y}$
            $G \leftarrow G + \frac{\delta C_m}{\delta Y}$
        **end**
        Set $Y^{(t)} = Y^{(t-1)} + \eta G + \alpha(t)(Y^{(t-1)} - Y^{(t-2)})$
    **end**
**end**

**Algorithm 1:** Multi-SNE

**Data:** $M$ data sets, $X^{(m)} \in \mathbb{R}^{N \times p_m}, \quad \forall m \in \{1, \cdots, M\}$
**Parameters:** $k$ [Number of neighbours]
**Result:** Induced embedding, $\hat{Y} \in \mathbb{R}^{n \times d}$. Often, $d = 2$
**begin**
    **for** *m=1 **to** M* **do**
        Find $k$ nearest neighbours of $X^{(m)}$.
        Compute $W^m$ by minimising equation (7).
    **end**
    Let $\hat{W} = \sum_m \alpha^m W^m$, where $\sum_m \alpha^m = 1$.
    Compute the $d$-dimensional embeddings $\hat{Y}$ by minimising equation (8) under $\hat{W}$.
**end**

**Algorithm 2:** multi-LLE

**Data:** $M$ data sets, $X^{(m)} \in \mathbb{R}^{N \times p_m}, \quad \forall m \in \{1, \cdots, M\}$
**Parameters:** $k$ [Number of neighbours]
**Result:** Induced embedding, $\hat{Y} \in \mathbb{R}^{n \times d}$. Often, $d = 2$
**begin**

    **for** $m=1$ **to** $M$ **do**

        Construct a $N \times N$ neighborouhood graph, $G_m \sim (V, E_m)$ with samples represented by nodes.
        The edge length between $k$ nearest neighbours of each node is measured by Euclidean distance.

    **end**

    Measure the average edge length between all nodes.
    Combine all neighborouhood graphs into a single graph, $\tilde{G}$.
    In $D_G \in \mathbb{R}^{|V| \times |V|}$, the computed shortest path distances between nodes in $\tilde{G}$ are stored.
    Compute the $d$-dimensional embeddings $Y$ by computing $y_i = \sqrt{\lambda_p} u_p^i$, where $\lambda_p$ is the $p^{th}$
     eigenvalue in decreasing order of the the matrix $\tau(D_G)$ and $u_p^i$ the $i^{th}$ component of $p^{th}$
     eigenvector. The operator, $\tau$ is defined by $\tau(D) = -\frac{HSH}{2}$, where $S$ is the matrix of squared
     distances defined by $S_{ij} = D_{ij}^2$, and $H$ is defined by $H_{ij} = \delta_{ij} - \frac{1}{N}$.

**end**

**Algorithm 3:** multi-ISOMAP

# B  Data Clustering Evaluation Measures

Let $\mathbf{X} = \{X_1, \cdots, X_r\}$ be the true classes of the data and $\mathbf{Y} = \{Y_1, \cdots, Y_s\}$ the clusterings found on $N$ objects. In this study, we assume to know the number of clusters and thus set $r = s$. Let $n_{ij}$ be the number of objects in $X_i$ and $Y_j$. A contingency table is defined as shown in Table 6.

|  | $Y_1$ | $Y_2$ | $\cdots$ | $Y_s$ | $\sum_j^s Y_j$ |
|---|---|---|---|---|---|
| $X_1$ | $n_{11}$ | $n_{12}$ | $\cdots$ | $n_{1s}$ | $\sum_i^r n_{1i} = a_1$ |
| $X_2$ | $n_{21}$ | $n_{22}$ | $\cdots$ | $n_{2s}$ | $a_2$ |
| $\vdots$ | $\vdots$ | $\ddots$ | $\vdots$ | $\vdots$ | $\vdots$ |
| $X_r$ | $n_{r1}$ | $n_{r2}$ | $\cdots$ | $n_{rs}$ | $a_r$ |
|  | $\sum_i^r n_{i1} = b_1$ | $b_2$ | $\cdots$ | $b_s$ | $\mathbf{S} = \sum_i^r \sum_j^s n_{ij}$ |

Table 6: A contingency table for data clustering. $X_i$ refers to the $i^{th}$ class (truth) and $Y_j$ refers to the $j^{th}$ cluster. $n_{ij}$ are the number of samples found in class $i$ and cluster $j$. In this study, $r = s$ was taken, as $K$-means with the true number of classes known was performed.

The formulas of the four measures used to evaluate data clustering are given below, with the terms defined in Table 6.

**Accuracy (ACC)**

$$(Acc) = \frac{\sum_i \sum_j \mathbb{1}\{i = j\} n_{ij}}{\mathbf{S}} \tag{12}$$

**Normalised Mutual Information (NMI)**

$$(NMI) = \frac{2I(\mathbf{X}, \mathbf{Y})}{H(\mathbf{X}) + H(\mathbf{Y})} \tag{13}$$

where $I(\mathbf{X}, \mathbf{Y})$ is the mutual information between $\mathbf{X}$ and $\mathbf{Y}$, and $H(\mathbf{X})$ is the entropy of $\mathbf{X}$.

**Rand Index (RI)**

$$(RI) = \frac{\binom{N}{2} - \left[\frac{1}{2}\sum_i(\sum_j n_{ij})^2 + \sum_j(\sum_i n_{ij})^2 - \sum_i\sum_j n_{ij}^2\right]}{\binom{N}{2}}$$

$$= \frac{\alpha + \beta}{\binom{N}{2}} \tag{14}$$

where $\alpha$ refers to the number of elements that are in the same subset in $X$ and in the same subset in $Y$, while $\beta$ is the number of elements that are in different subsets in $X$ and in different subsets in $Y$.

**Adjusted Rand Index (ARI)**

$$(ARI) = \frac{\sum_i \sum_j \binom{n_{ij}}{2} - \frac{\sum_i \binom{a_i}{2} \sum_j \binom{b_j}{2}}{\binom{n}{2}}}{\frac{1}{2}\left[\sum_i \binom{a_i}{2} + \sum_j \binom{b_j}{2}\right] - \frac{\sum_i \binom{a_i}{2} \sum_j \binom{b_j}{2}}{\binom{n}{2}}} \tag{15}$$

## C  Single-cell data

In the multi-omics single-cell data analysis, we used the publicly available data set provided by 10x Genomics for human peripheral blood mononuclear cells (PBMC) [6]. This data set can be downloaded and installed via the R package `SeuratData`, by running the command `InstallData("pbmcMultiome")`. In their vignette [7], Hoffman et al. (2021) explored this data set to demonstrate how to jointly integrate and analyse such data.

In this data set, scRNA-seq and scATAC-seq profiles were simultaneously collected in the same cells by 10x Genomics. Data on 11909 single cells are available on 36601 genes and 108377 peaks in scRNA-seq and scATAC-seq, respectively. Cells with zero summed expression, along all genes were removed, leaving us with 10412 cells. Pre-processing was employed via the `Seurat` package, following the steps performed by Hoffman et al. (2021). Firstly, we log-normalised both data-views and then selected features for each individual data-view. In feature selection, we aim to identify a subset of features with high variability across cells (using the functions `FindVariableFeatures` and `FindTopFeatures`) (Stuart et al., 2019).

The multi-omics single-cell data set consists of 19 imbalanced clusters that correspond to their corresponding cell types; we assume the annotations provided by `Seurat` to be accurate (Figure 13). To evaluate the clustering performance of multi-SNE, we took a balanced subset of the data. Cell-type clusters with less than 200 cells were removed entirely and we combined cells with cell types under the same hierarchy. For example, *Intermediate B*, *Naive B* and *Memory B* were combined to create a single cluster, *B cells*. Similarly, *CD4 Naive*, *CD4 TCM* and *CD4 TEM* were combined as *CD4 cells*. After this process, we ended up with a subset of 9105 single cells separated in 6 cell-type clusters (*B cell*, *CD14 Mono*, *CD4*, *CD8 Naive*, *CD8 TEM* and *NK*).

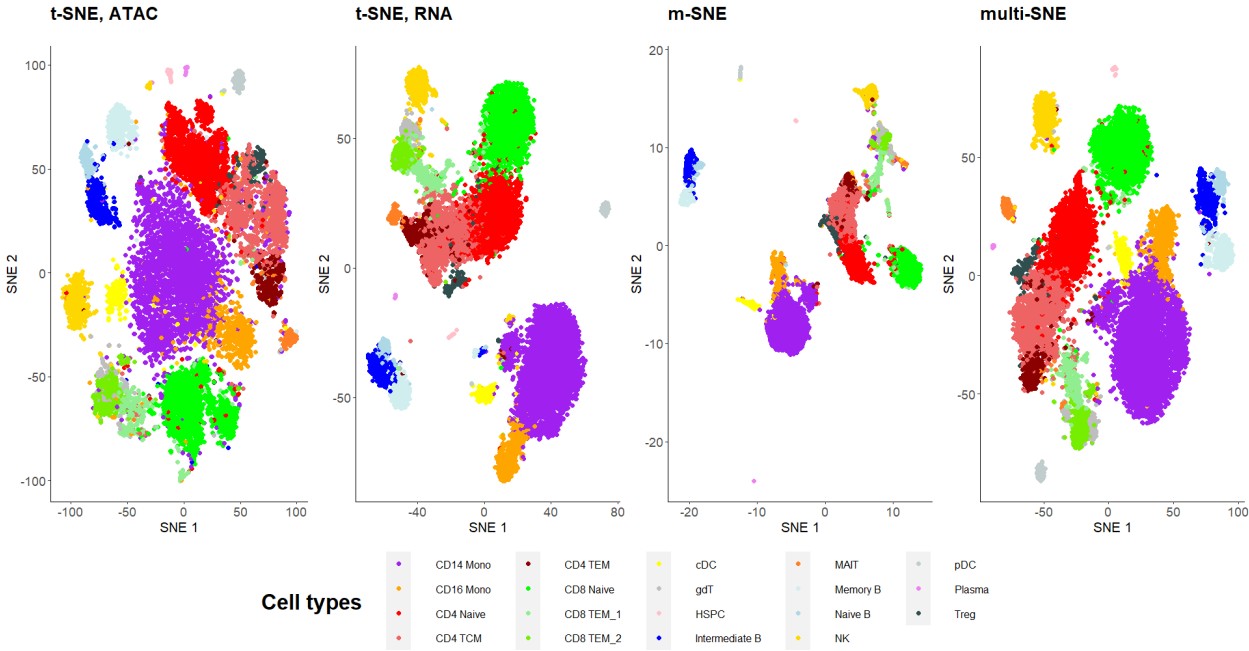

Figure 13: **Visualisations of single-cell data.** Projections of the full data set with unbalanced clusters produced by t-SNE on RNA, ATAC, m-SNE and multi-SNE on both data-views with perplexity $Perp = 80$ for the two t-SNE projections, $Perp = 100$ for m-SNE and $Perp = 20$ for multi-SNE.

M-SNE and multi-SNE combined the scRNA-seq and scATAC-seq to produce a more intelligible projection of the cells than t-SNE applied on either data-view. Qualitatively the superiority of the multi-view manifold learning algorithms may not be obvious at first, but subtle differences can be observed. Quantitatively,

---

[6] `https://support.10xgenomics.com/single-cell-multiome-atac-gex/datasets/1.0.0/pbmc_granulocyte_sorted_10k`

[7] `https://satijalab.org/seurat/articles/atacseq_integration_vignette.html`

multi-SNE received the best evaluation scores, with $NMI = 0.807$, while m-SNE received $NMI = 0.760$. Single-view t-SNE scored $NMI = 0.620$ and $NMI = 0.572$ for scRNA-seq and scATAC-seq, respectively.

# D   Additional comparisons

## D.1   Multi-SNE, m-SNE and MV-tSNE2

This section justifies the exclusion of MV-tSNE2 from the comparisons against multi-SNE. Due to its superior performance, m-SNE was selected as an existing competitor of multi-SNE.

Multi-SNE and m-SNE outperformed MV-tSNE2 on all data sets presented in this manuscript (Figure 14). By comparing the produced visualisations on two data sets, Figure 14 evaluates the three algorithms qualitatively. Multi-SNE produced the best separation among clusters on both data sets. In MV-tSNE2, a lot of the samples are projected bundled together, making it difficult to distinguish the true clusters. Quantitative evaluation of the methods agree with the conclusions reached by assessing the visualisations qualitatively.

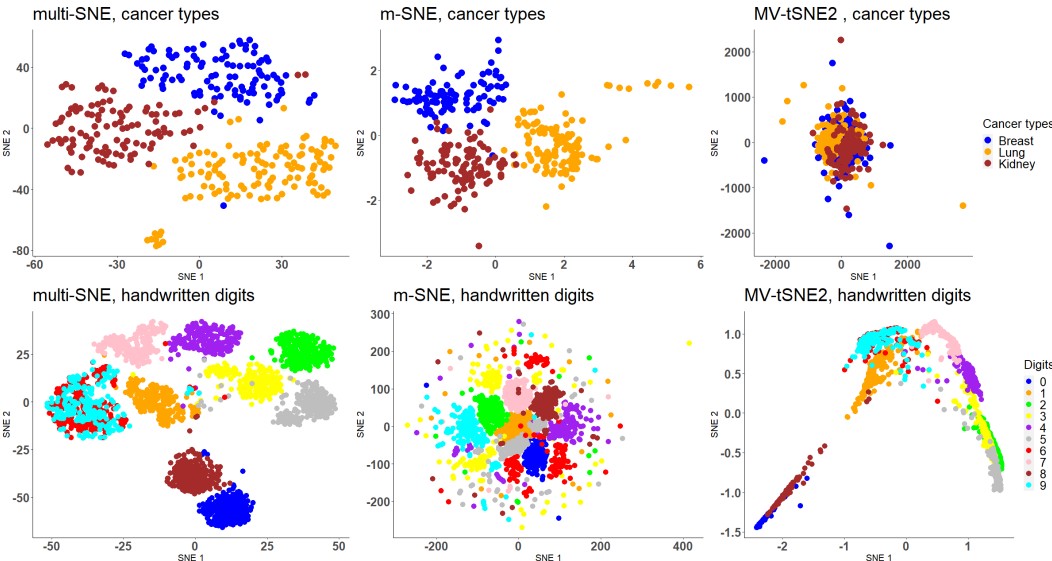

Figure 14: **Visualisations by multi-SNE, m-SNE and MV-tSNE2.** The three multi-view SNE-based projections of cancer types and handwritten digits data sets.

## D.2 Multi-SNE, j-SNE and j-UMAP

At the same time as multi-SNE was developed, Hoan Do & Canzar (2021) proposed generalisations of t-SNE (named j-SNE) and UMAP (named j-UMAP) based on a similar objective function as multi-SNE. Hoan Do & Canzar (2021) introduced a regularisation term that reduces the bias towards specific data-views; the proposed objective function is given by:

$$C_{j-SNE} = \sum_m \sum_i \sum_j \alpha^m p_{ij}^m \log \frac{p_{ij}^m}{q_{ij}} + \lambda \sum_m \alpha^m \log \alpha^m, \tag{16}$$

where $\alpha^m$ represents the weight provided for the $m^{th}$ data-view and $\lambda$ is a regularisation parameter. The weights and low-dimensional embeddings are updated iteratively. The adjustments on the weights of each data-view are performed in accordance to the regularisation parameter, which requires tuning for optimal results.

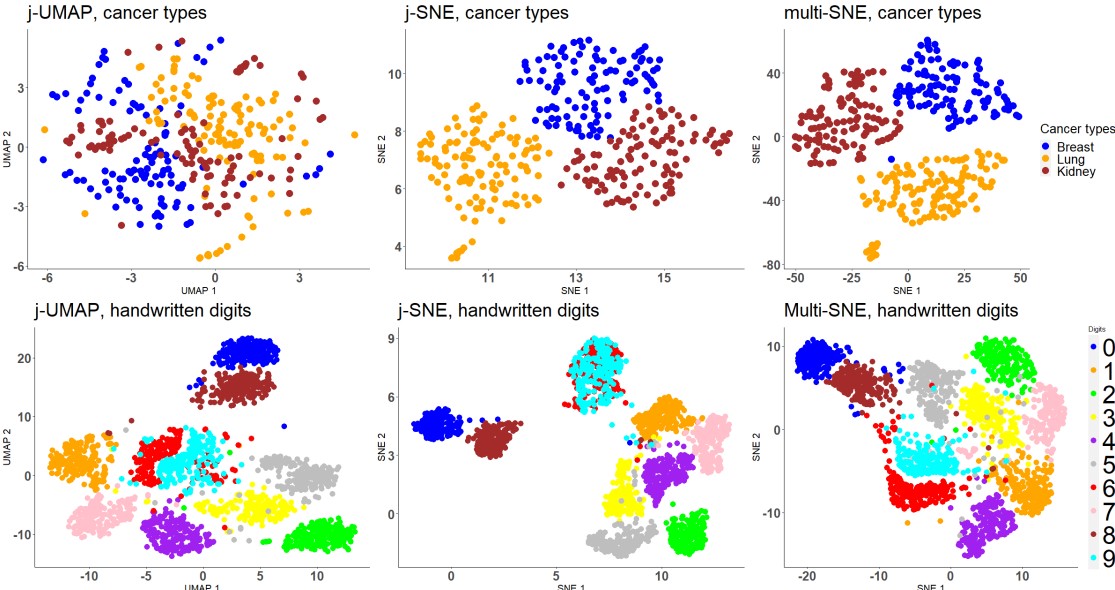

Figure 15: **j-UMAP, j-SNE and multi-SNE visualisations.** Projections of cancer types and handwritten digits data sets, produced by j-UMAP, j-SNE and multi-SNE.

Figure 15 compares qualitatively multi-SNE with j-SNE and j-UMAP (with their respective tuning parameters optimised) on the cancer types and handwritten digits data. As expected, the projections by j-SNE and multi-SNE are very much alike for both data sets. The increasing complexity imposed by the regularisation term in j-SNE does not seem to benefit the visualisation of the samples. j-UMAP does not separate the three cancer types, but it manages to separate the 10 digits, even samples that represent the 6 and 9 numerals; j-SNE failed to do that. This was achieved by multi-SNE at the 3-dimensional visualisation, or alternatively by using multi-CCA as a pre-training step. All three algorithms allocated similar weight values to each data-view on both data sets. In particular, transcriptomics on cancer types and morphological features on handwritten digits received the lowest weight.

### D.3 Tuning parameters on real data

In this section, we have explored how the parameter values affect the multi-view approaches when analysing real data sets. Figure 16 depicts the NMI evaluation measure on each real data set for parameter values in the range $S$.

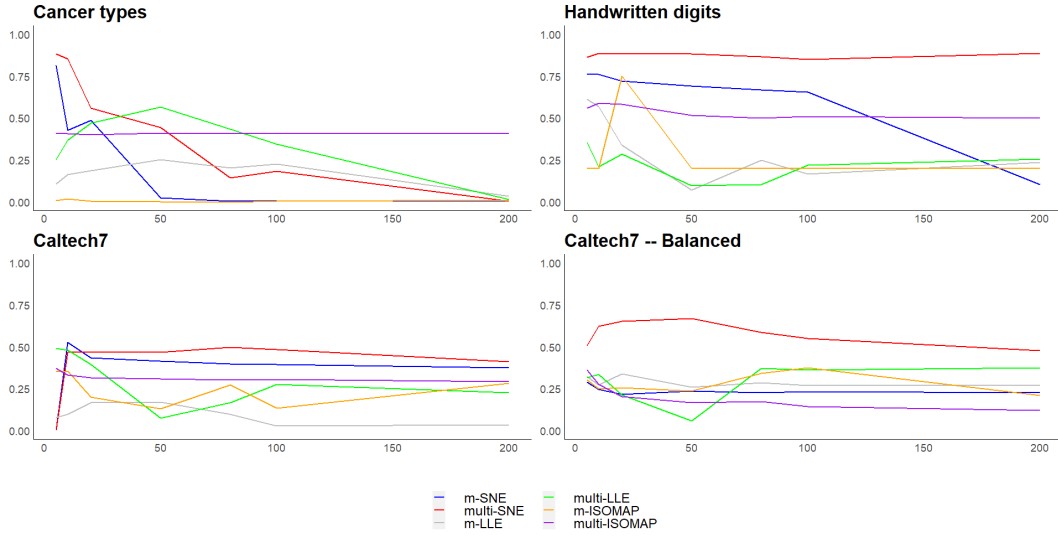

Figure 16: **Real data sets evaluation via NMI**. The NMI values are plotted against different parameter values on all multi-view manifold learning algorithms investigated in this manuscript.

Similar conclusions to the ones made in Section 4.3 were reached (Figure 16). SNE-based solutions had a more consistent performance than LLE and ISOMAP based approaches. In contrast to the conclusions reached by testing the tuning parameters on synthetic data, SNE-based approaches applied to the cancer types data set, performed the best when the perplexity was low. This observation highlights the importance of the tuning parameters (perplexity and the number of neighbours) in these algorithms, as discussed by their respective authors. For the remaining data sets, its performance was stable on different parameter values. With the exception of cancer types data, the performance of LLE-based solutions form similar behaviour with the synthetic data (*i.e.* their performance is reduced around $NN = 50$ and then it is regained.

### D.4 Increased randomness in data

We have further explored how additional noise affects the performance of the multi-view learning approaches.

As discussed, the NDS data set contains three informative data-views and one noisy data-view. In Section 4.4, we concluded that the inclusion of the noisy data-view reduces the performance both qualitatively and quantitatively. This complication was targeted and solved through an automatic weight-updating approach in Section 4.5.1. The purpose of this section is to test the performance of multi-view manifold learning solutions on data sets with higher levels of randomness. To increase the noise in the synthetic data, additional noisy data-views were generated. In particular, this section compares the performance of manifold learning algorithms on three synthetic data sets: (a) NDS, (b) NDS with one additional noisy data-view, and (c) NDS with two additional noisy data-views. Each simulation was performed for 200 runs and with equal weights for a fair comparison.

Multi-SNE was the superior algorithm in all simulations (Table 7). With each additional noisy data-view, all multi-view manifold learning algorithms saw a reduction in their performance. Although all evaluation measures reflect this observation, the change in their performance is best observed on the NMI values (Table 7). Further, with more noisy data-views, the variance of the evaluation measures increased. This observation suggests that all algorithms clustered the samples with higher uncertainty.

The proposed automatic weight-adjusting process ensures that all data-views receive a weight value, which suggests that noisy data-views do not receive zero weight. For example, in the NDS with two additional noisy data-views scenario, this process returned higher weights for the informative data-views ($X1$, $X2$, $X3$), than the noisy ones ($X4$, $X5$, $X6$) (Figure 17). Although the weights between informative and noisy data-views are close in value, the proposed automatic weight-adjusting process can successfully distinguish informative from noisy data-views (Figures 17). To further assess if this process allocates substantial weight to noisy data-views, a scenario in which the data set contains more noisy data-views than informative ones was investigated.

In particular, a simulation was performed in which 1 data-view is informative and 2 are noisy. The informative data-view contains information to split the samples into 3 clusters, while the 2 noisy data-views assign all samples on the same cluster. Multi-SNE separates the samples by their respective cluster, despite having more noisy data-views than informative ones (Figure 18). In accordance with the other simulations, multi-SNE assigns a higher weight value on the informative data-view than on the noisy data-views. The informative data-view received a weight of 0.4 and each of the two noisy data-view 0.3 (Figure 18). This difference in the weights between the data-views acts as an incentive for the user to investigate the implementation of the algorithm by excluding the data-view(s) that received the lowest weight(s).

| Data Set | Algorithm | Accuracy | NMI | RI | ARI |
|---|---|---|---|---|---|
| | SNE$_{concat}$ [Perp=80] | 0.747 | 0.628 | 0.817 | 0.598 |
| | m-SNE [Perp=50] | 0.650 | 0.748 | 0.766 | 0.629 |
| | **multi-SNE** [Perp=80] | 0.989 | 0.951 | 0.969 | 0.987 |
| | LLE$_{concat}$ [NN=5] | 0.606 | 0.477 | 0.684 | 0.446 |
| NDS | m-LLE [NN=20] | 0.685 | 0.555 | 0.768 | 0.528 |
| | multi-LLE [NN=20] | 0.937 | 0.768 | 0.922 | 0.823 |
| | ISOMAP$_{concat}$ [NN=100] | 0.649 | 0.528 | 0.750 | 0.475 |
| | m-ISOMAP [NN=5] | 0.610 | 0.453 | 0.760 | 0.386 |
| | multi-ISOMAP [NN=300] | 0.778 | 0.788 | 0.867 | 0.730 |
| | SNE$_{concat}$ [Perp=80] | 0.723 | 0.648 | 0.787 | 0.585 |
| | m-SNE [Perp=50] | 0.623 | 0.705 | 0.734 | 0.605 |
| | **multi-SNE** [Perp=80] | 0.983 | 0.937 | 0.951 | 0.966 |
| | LLE$_{concat}$ [NN=5] | 0.575 | 0.427 | 0.628 | 0.402 |
| Higher dimension | m-LLE [NN=20] | 0.671 | 0.534 | 0.755 | 0.513 |
| | multi-LLE [NN=20] | 0.903 | 0.788 | 0.898 | 0.802 |
| | ISOMAP$_{concat}$ [NN=100] | 0.622 | 0.510 | 0.705 | 0.453 |
| | m-ISOMAP [NN=5] | 0.589 | 0.439 | 0.734 | 0.344 |
| | multi-ISOMAP [NN=300] | 0.765 | 0.767 | 0.859 | 0.711 |
| | SNE$_{concat}$ [Perp=10] | 0.650 | 0.522 | 0.724 | 0.489 |
| | m-SNE [Perp=100] | 0.689 | 0.584 | 0.786 | 0.530 |
| | **multi-SNE** [Perp=50] | 0.965 | 0.854 | 0.956 | 0.901 |
| | LLE$_{concat}$ [NN=10] | 0.604 | 0.445 | 0.723 | 0.413 |
| One additional noisy | m-LLE [NN=10] | 0.667 | 0.522 | 0.765 | 0.490 |
| | multi-LLE [NN=5] | 0.912 | 0.692 | 0.891 | 0.756 |
| | ISOMAP$_{concat}$ [NN=20] | 0.543 | 0.375 | 0.733 | 0.481 |
| | m-ISOMAP [NN=20] | 0.552 | 0.482 | 0.703 | 0.444 |
| | multi-ISOMAP [NN=5] | 0.584 | 0.501 | 0.739 | 0.493 |
| | SNE$_{concat}$ [Perp=10] | 0.581 | 0.310 | 0.688 | 0.309 |
| | m-SNE [Perp=10] | 0.603 | 0.388 | 0.712 | 0.359 |
| | **multi-SNE** [Perp=10] | 0.936 | 0.781 | 0.926 | 0.832 |
| | LLE$_{concat}$ [NN=10] | 0.523 | 0.251 | 0.641 | 0.222 |
| Two additional noisy | m-LLE [NN=10] | 0.570 | 0.344 | 0.682 | 0.317 |
| | multi-LLE [NN=5] | 0.858 | 0.557 | 0.832 | 0.622 |
| | ISOMAP$_{concat}$ [NN=20] | 0.470 | 0.389 | 0.565 | 0.409 |
| | m-ISOMAP [NN=20] | 0.489 | 0.406 | 0.611 | 0.453 |
| | multi-ISOMAP [NN=5] | 0.524 | 0.467 | 0.782 | 0.517 |

Table 7: **Clustering performance of NDS and with additional noisy data-views.** For each data set, red highlights the method with the best performance on each measure between each group of algorithms (SNE, LLE or ISOMAP based). The overall superior method for each data set is depicted with **bold**. The parameters *Perp* and *NN* refer to the selected perplexity and number of nearest neighbours, respectively. They were optimised for the corresponding methods.

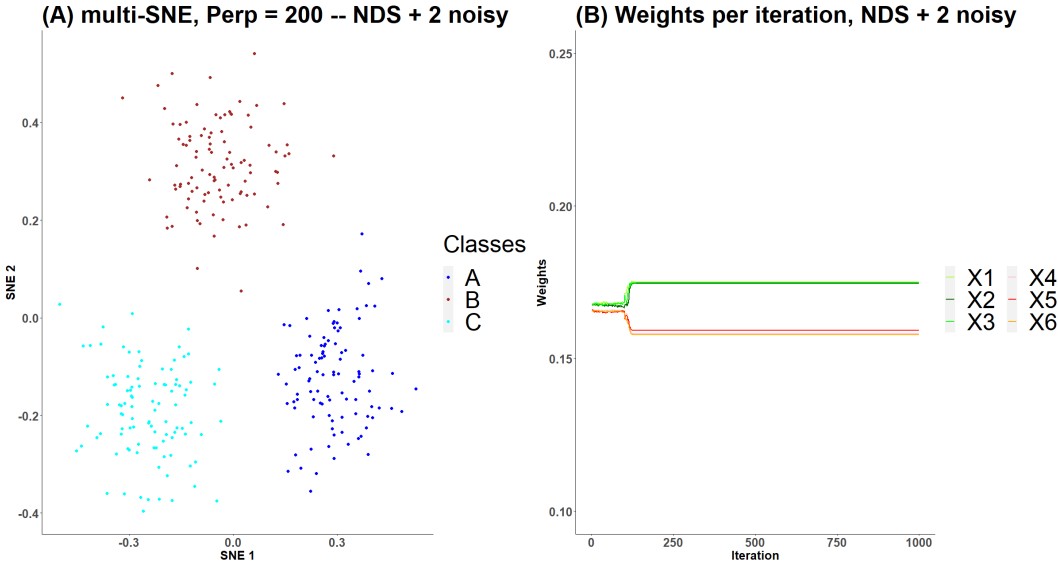

Figure 17: **Visualisations on NDS with** 2 **additional noisy data-views.** (A) Scatter-plot of the simulated samples obtained by multi-SNE with perplexity $Perp = 200$ and (B) Weights received by the algorithm on the 6 data-views. The first 4 follow the structure of NDS simulation; three informative data-views and a noisy one. Each informative data-view separates the samples differently, but taken together they are split equally into three clusters. $X5$ and $X6$ represent the 2 additional noisy data-views.

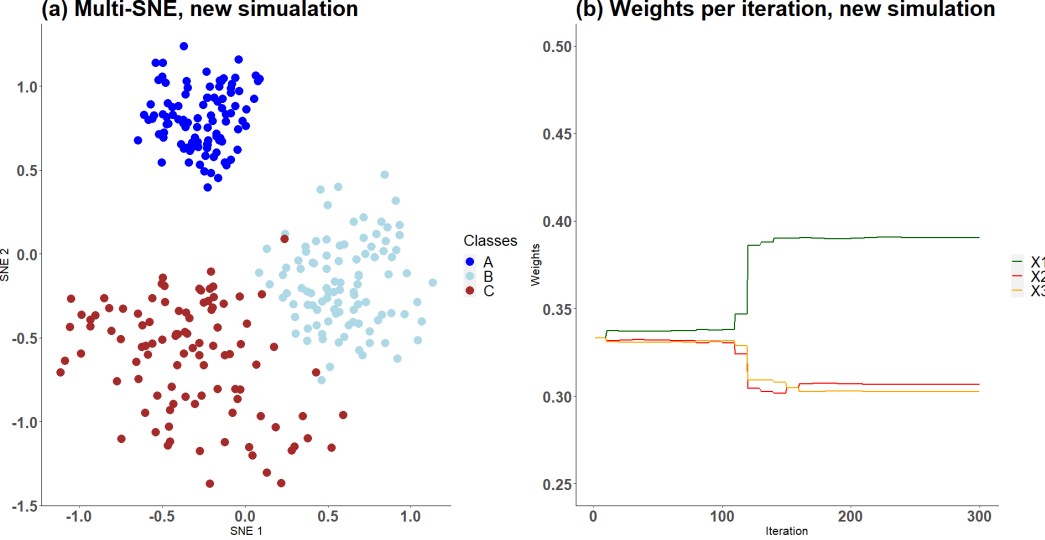

Figure 18: **Visualisations on new simulation with** 1 **informative and** 2 **noisy data-views.** (A) Scatter-plot of the simulated samples obtained by multi-SNE with perplexity $Perp = 100$ and (B) Weights received by the algorithm on the 3 data-views. The informative data-view contains information to split the samples into 3 clusters, while the 2 noisy data-views assign all samples to lie on the same cluster.

### D.5 t-SNE on single-view cancer types

Table 8 presents the clustering performance of t-SNE applied on the three views in the cancer types data set, separately. Genomics was the favoured view on all evaluation measures.

| | ACC | NMI | RI | ARI |
|---|---|---|---|---|
| Genomics | 0.595 (0.044) | 0.299 (0.041) | 0.667 (0.017) | 0.253 (0.039) |
| Epigenomics | 0.500 (0.036) | 0.116 (0.033) | 0.598 (0.018) | 0.107 (0.035) |
| Transcriptomics | 0.456 (0.023) | 0.042 (0.011) | 0.572 (0.006) | 0.049 (0.013) |

Table 8: Clustering performance on the induced embedding of a single view obtained by implementing t-SNE on Cancer Types data. Standard deviation is reported in parentheses.

### D.6 Handwritten digits projection in 3 dimensions (3D)

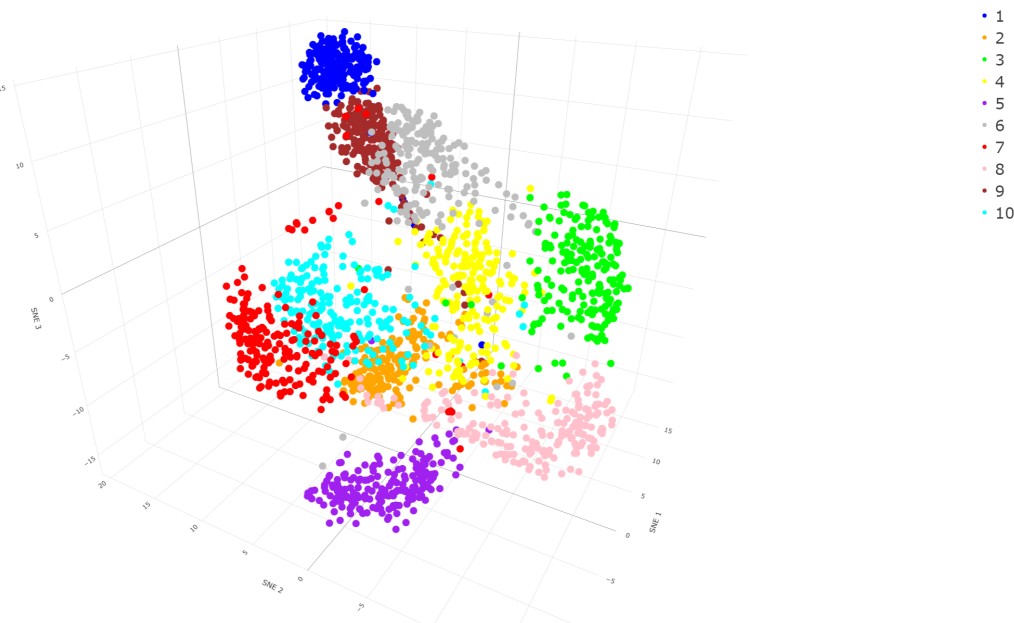

Figure 19: **3D multi-SNE visualisation of handwritten digits**. Projections produced by weight adjusting multi-SNE with multi-CCA as pre-training and perplexity $Perp = 80$. Colours present the true clustering of the data points.

### D.7 Alternative quantitative evaluation measures for clustering

Accuracy (ACC), Normalised Mutual Information (NMI), Rand Index (RI) and Adjusted Rand Index (ARI) are the evaluation measures chosen to quantitatively evaluate the clustering performance of the proposed multi-view approaches. These measures were chosen because the true annotations of the data sets are known and together they provide a wide assessment range. Practically, clustering is often applied to data with unknown annotations (labelling), therefore for completeness, we have further explored the implementation of the Silhouette score for identifying the optimal tuning parameter of the manifold visualisation approaches. The Silhouette score is a widely used measure for quantifying the clustering produced by the clustering algorithms, or for selecting the optimal number of clusters.

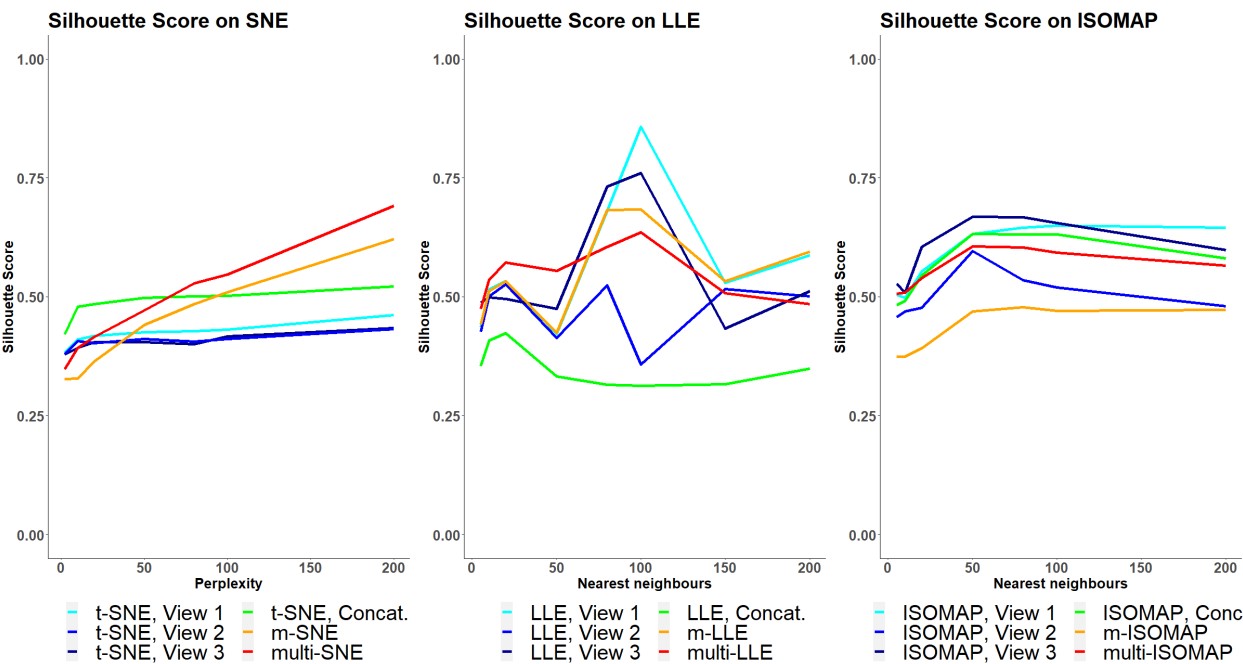

Figure 20: **Silhouette score on MCS.** The clustering evaluation via Silhouette score is plotted against different parameter values on all SNE, LLE and ISOMAP based algorithms.

Figure 20 presents the evaluation performance of the methods with respect to their tuning parameter when the Silhouette score is evaluated instead of the other four measures. This figure complements Figure 7. The Silhouette score is not always in agreement with the other evaluation measures. For example, in SNE-based solutions, according to the Silhouette score, multi-SNE is favoured over the other methods only when perplexity is 100. Another difference between the silhouette score and other measures is that as the perplexity increases, multi-SNE remains stable for the other measures (for $Perp \geq 50$), while its silhouette score keeps increasing. Silhouette score measures how well the clusters are separated between them. This is conceptually different from what the other measures quantify which is how well the proposed clusters agree with the known clusters. It is therefore of no surprise that the findings are not always in agreement.

## D.8 Alternative clustering algorithms

For the clustering task of the samples, any clustering algorithm could have potentially been applied to the low-dimensional embeddings produced by the multi-view visualisation approaches proposed.

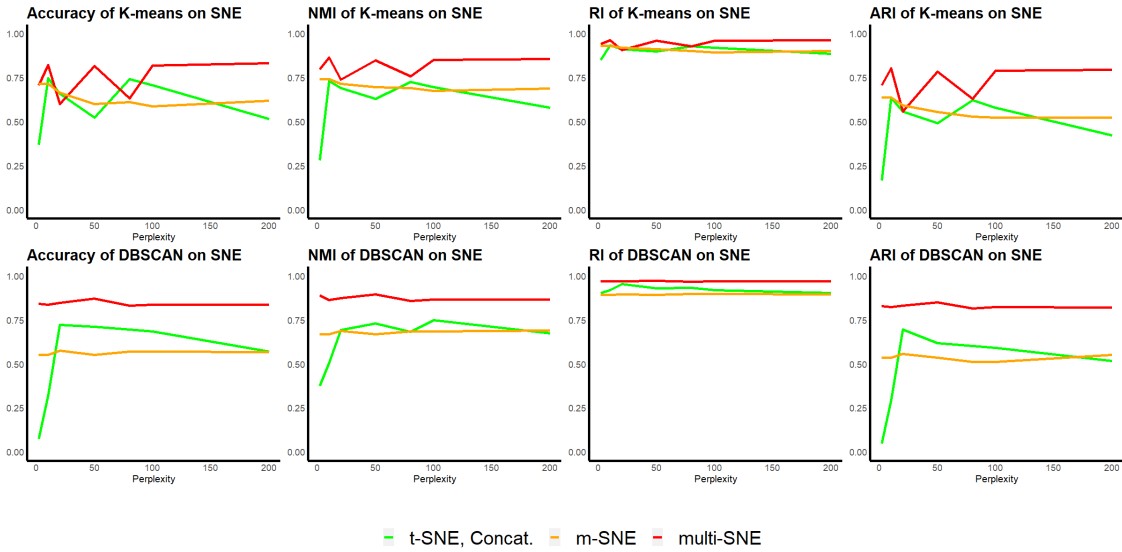

Figure 21: **$K$-means and DBSCAN on SNE-based solutions applied on handwritten digits data set.** The clustering evaluation measures are plotted against different perplexity values on multi-SNE, m-SNE and $SNE_{concat}$. The performance of $K$-means and DBSCAN applied on the produced embeddings is depicted in the first and second row of this figure, respectively

In the main body of this manuscript, the $K$-means algorithm was chosen due to its popularity, its strong robust performance and because the true number of clusters is known for all data sets. In practice, the latter is not always true, and clustering algorithms that do not require the number of clusters as a parameter input are preferable. Density-based spatial clustering of applications with noise (DBSCAN) is an example of such an algorithm (Ester et al., 1996). DBSCAN instead requires two other tuning parameters: the minimum number of samples required to form a dense cluster and a threshold in determining the neighbourhood of a sample, named $\epsilon$.

The implementation of DBSCAN on handwritten digits, smooths the performance of SNE-based solutions across different perplexity values (Figure 21). For all parameter values, DBSCAN performs equally well, while the performance of $K$-means slightly oscillates.

A greater disagreement between the two unsupervised learning algorithms is observed in their application to caltech7 data set (Figure 22). While the accuracy of multi-SNE by implementing $K$-means reduces with higher perplexity, the opposite behaviour is observed when DBSCAN is implemented. In addition, DBSCAN finds multi-SNE to be superior over m-SNE, while $K$-means concludes the opposite.

This appendix demonstrates that the implementation of clustering on the produced embeddings is not restricted only to $K$-means, but alternative clustering solutions may be used. In particular, DBSCAN is a good choice, especially when the true number of clusters is unknown.

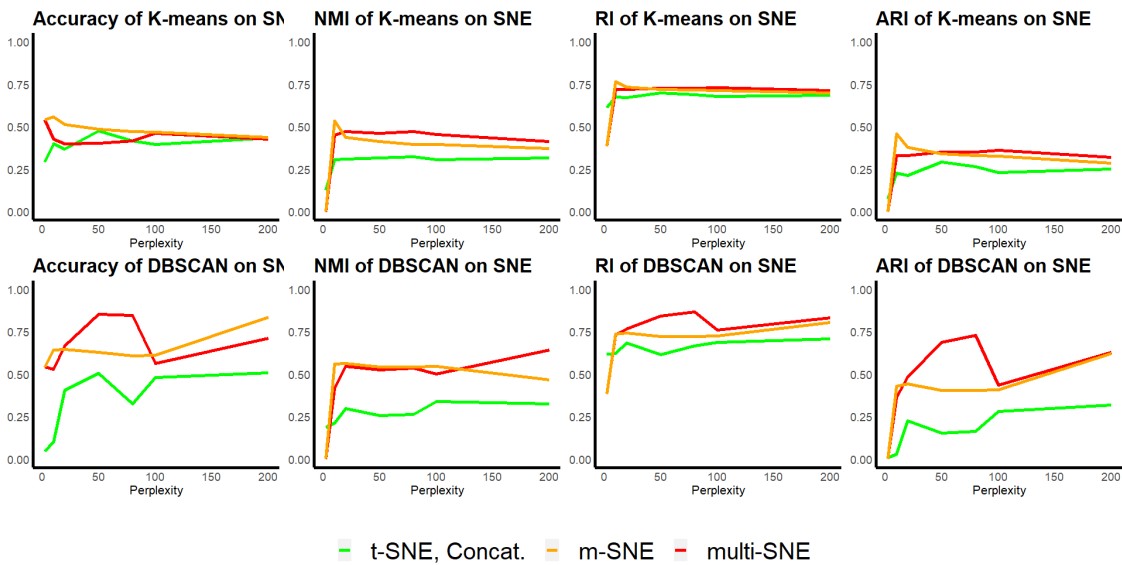

Figure 22: **$K$-means and DBSCAN on SNE-based solutions applied on caltech7 data set.** The clustering evaluation measures are plotted against different perplexity values on multi-SNE, m-SNE and $SNE_{concat}$. The performance of $K$-means and DBSCAN applied on the produced embeddings is depicted in the first and second row of this figure, respectively

# E    Reproducibility

The code of multi-SNE was based on the publicly available software, written by the author of t-SNE, found in the following link:
`https://lvdmaaten.github.io/tsne/`

In this manuscript, all t-SNE results were obtained by running the original `R` implementation (https://cran.r-project.org/web/packages/tsne/) and verified by the original `Python` implementation (`https://lvdmaaten.github.io/tsne/`).

We refer the readers to follow the code and functions provided in the link below to reproduce the findings of this paper. The software for m-SNE and m-LLE were not found publicly available and thus we used our own implementation of the method that can be found in the same link below:
`https://github.com/theorod93/multiView_manifoldLearning`

An `R` package that contains the code for multi-SNE can be installed via `devtools` and it can be found in `https://github.com/theorod93/multiSNE`

We refer the readers to follow the links provided in the main body of the paper for the public multi-view data used in this paper.

# F  Computation time

In terms of computation time, none of the multi-view manifold learning algorithms was consistently faster than the rest (Table 9). However, multi-SNE was often the slowest algorithm, while m-SNE and multi-ISOMAP had the fastest computation time. The SNE-based solutions are based on the original t-SNE algorithm, as described in Appendix E.

**Running time**

| | MMDS | NDS | MCS | Caltech7 | Handwritten Digits | Cancer Types |
|---|---|---|---|---|---|---|
| m-SNE | 0.43 (0.019) | 0.29 (0.07) | 0.42 (0.01) | 4.29 (0.54) | 13.34 (3.43) | 251.71 (15.68) |
| multi-SNE | 1.07 (0.100) | 0.78 (0.14) | 1.02 (0.01) | 15.95 (0.71) | 45.76 (8.44) | 252.00 (11.23) |
| m-LLE | 0.25 (0.071) | 0.40 (0.12) | 0.42 (0.34) | 37.5 (2.21) | 26.28 (8.82) | 159.52 (17.49) |
| multi-LLE | 0.28 (0.099) | 0.41 (0.15) | 0.30 (0.14) | 37.9 (2.57) | 27.94 (5.29) | 157.73 (18.19) |
| m-ISOMAP | 0.22 (0.015) | 0.57 (0.06) | 0.37 (0.01) | 38.07 (3.09) | 29.52 (5.37) | 154.83 (18.04) |
| multi-ISOMAP | 0.24 (0.032) | 0.54 (0.13) | 0.33 (0.05) | 21.23 (2.24) | 16.77 (4.65) | 85.47 (14.57) |

Table 9: **Averaged running time recorded in minutes.** Taken for each manifold learning algorithm on all data sets seen in this paper; standard deviation is given in parentheses. All algorithms ran on High Performance Computing with 4 nodes.

