# OpenReview forum: "Multi-view Data Visualisation via Manifold Learning"
_TMLR — Rejected by TMLR_

### Review · Reviewer_gKV1 · 2023-07-24

**Summary Of Contributions:**

In their paper "Multi-view data visualization via manifold learning" the authors suggest a generalization of t-SNE (as well as LLE and ISOMAP) to multi-view data, i.e. datasets where several distinct sets of features are available for the same set of samples. They show that their suggested "multi-SNE" performs better than naive approaches like feature concatenation, and better than competitor methods, on toy simulations and some real-world datasets.

**Audience:**

Yes

**Broader Impact Concerns:**

No concerns.

**Claims And Evidence:**

Yes

**Requested Changes:**

All "major issues" are critical to securing my recommendation, but should be all straightforward to implement.

MAJOR ISSUES:

* The j-SNE paper is referenced only briefly in the Introduction, and then only in the Appendix the authors say that it was parallel work and that the loss function is nearly identical. I was surprised to see that it was parallel work, given that the j-SNE paper was published in 2021. So I checked the arXiv versions of both papers and realized that they both appeared in January 2021 for the first time. I think the authors need to be *very explicit* about it. The Introduction should say that their method is very similar to the one suggested in the j-SNE paper but that it was parallel work -- and explicitly say that both papers appeared on arXiv in Jan 2021 for the first time. I think this is very important, to set the record straight.

* In Section 4.1: "... because that data-view has a higher variability between the clusters than the other two". This is a crucial aspect of the simulation, but it was **not described** in Section 3.1, or at least I cannot find it there. Please explain this crucial simulation setup in Section 3.1.

* Section 4.1: Of course feature concatenation will fail if one data-view has a lot higher variance than another data-view. This can to some extent be remedied by normalizing each data-view before concatenation, for example by dividing each data-view matrix by its Frobenius norm. This would be less of a strawman comparison, and I suggest the authors do something like that.

* Some font sizes in Figures 6 and 7 are too small and unreadable when the paper is printed out.

* Discussion, 2nd paragraph, and Table 9: Unclear what t-SNE approximation was used. Was it vanilla t-SNE (which is O(n^2)), was it Barnes-Hut t-SNE, or something else? Please clarify somewhere the implementation details. If it's vanilla t-SNE, please mention **explicitly** that the implementation as you provide it, cannot be run for large sample sizes (larger than ~10,000). This is a very serious limitation that needs to be explicitly stated. Also specify that your software is in R. All of this should be in the main text, not in the Appendix.


MINOR ISSUES:

* ">>" in formulas should be typeset as \gg

* The j-SNE paper has the authors swapped for some reason. Canzar is the second author: https://genomebiology.biomedcentral.com/articles/10.1186/s13059-021-02356-5, not the first.

* Last paragraph in Section 2.3: What is a "consensus matrix" here exactly?

* Step in Section 2.4: should it be d_x = \infty if ij is not in the kNN graph? Instead of d_x = 0.

* Step 3 in Section 2.4: I don't think that's how ISOMAP works. ISOMAP applies "classical MDS" to the distance matrix D, which means that it double-centers it before computing the eigendecomposition.

* Section 3.1: the description is confusing because it contains spherical Gaussian noise TWICE. First you sample from a Gaussian with mean \mu and spherical covariance I_p_m. And then you add a noise vector epsilon sampled from another Gaussian with mean 0 and covariance I_p_m (same!). Is this intentional? It's a very strange setup as it is equivalent to sampling once using larger variance. Or is it an inaccurate description?

* Table 1: Why is the NDS setup "high-dimensional"? Is it because p>n? Please clarify.

**Strengths And Weaknesses:**

STRENGTHS:

* The paper addresses an important problem, as multi-view data become more and more prevalent in application fields like single-cell biology where manifold learning is frequently used.
* The suggested methods (multi-SNE) is reasonable and straightforward, and performs well in practice.
* The authors provide an overview of multi-view manifold learning methods based on LLE/ISOMAP and suggest and test their own versions of multi-view LLE (multi-LLE) and multi-view ISOMAP (multi-ISOMAP).

WEAKNESSES:

* A very similar paper has been published two years ago in Genome Biology (https://genomebiology.biomedcentral.com/articles/10.1186/s13059-021-02356-5) suggesting multi-view versions of t-SNE and UMAP called j-SNE and j-UMAP. In fact, multi-SNE is very similar to j-SNE (and moreover, j-SNE is arguably a more flexible generalization as it allows automatic selection of non-equal modality weights). That said, I checked and confirmed that both papers first appeared on arXiv in January 2021, and so can be considered parallel work. However, the current paper needs to be more explicit about that, see below.

* The feature concatenation comparison is a bit of a strawman, as it does not perform any feature normalization (see below).

* While it is interesting to consider multi-view generalizations of LLE and ISOMAP, it is clear that for the considered toy examples and real-world datasets they will lose to multi-SNE, as LLE and ISOMAP perform much worse than t-SNE on such data. Indeed, the authors find that multi-SNE performs the best.

* Some of the methods description is not sufficiently clear (see below).

Overall I think the paper can be accepted to TMLR after some straightforward revision.

---

> ### Author Response · Authors · 2023-08-23
>
> We would like to thank you for your comments and recommendations for improving our manuscript. Below we have addressed the raised issues. We hope that you will find our updated manuscript in response to your comments suitable for publication.
>
> A. Parallel work
>
> Thank you. for taking the time to check the two manuscripts. As you have pointed out both manuscripts were submitted as preprints in January 2021. As you have recommended it is important to set the record straight and discuss that the two works were done in parallel. Following this, we have updated the Introduction to include this information for the readership.  A comparison of the proposed multi-SNE with j-SNE is included in Appendix D, where we found multi-SNE visualisations to outperform the ones produced by j-SNE.
>
> B. Simulation study design
>
> You are correct, our simulation strategy incorporates a noise parameter, $\epsilon$, which would be equivalent to sampling once from a Gaussian distribution with a larger variance. The reason we separated the two noises, was to explicitly define data-views with higher variability. In other words, this additional error was only added to some of the data-views included. The reason for this additional noise is to assess whether the algorithms would be able to recognise and accommodate a data-view with larger variability than the rest. Across all simulation scenarios a single data-view that was simulated with an additional noise, epsilon, was included. This two-step process allows for a simpler explanation and implementation of the simulation scenarios, by considering selected data-views with higher variability than the rest.
>
> The description of the simulation strategy has been updated to clearly explain the logic and the steps implemented in obtaining simulated data. In brief, an additional noise parameter, epsilon, was included in the simulation strategy to increase the variability of some of the data-views.
>
> The manuscript has been updated with further details about the role of epsilon in Sections 3.1, MMDS simulation, and MCS simulation.
>
> C. Data Concatenation
>
> We agree that a non-proper concatenation of the data can lead to spurious relationships, therefore before doing any analysis. In our conducted work, the data-views of both real and simulated data, were normalised before concatenation in every run. Thank you for pointing out that this important information was not included in the manuscript. We have therefore updated Section 4 to include that normalization of the data was performed before applying the manifold learning approaches. The normalisation of the data was done by removing the mean and dividing by the standard deviation of each feature in the data-views.
>
> D. T-SNE version
>
> The original (or vanilla) t-SNE was used to implement the algorithm. Multi-SNE is based on the same solution, and not on any other variation of t-SNE. We agree with the reviewer that t-SNE cannot run on large sample sizes, and we have therefore already pointed out at the end of Section 2.2 that we followed van der Maaten and Hinton (2008) and applied PCA as a pre-training step to our analyses. We have updated Section 2.2 and Appendix E to include information about the R and Python packages used for our multi-SNE implementation.
>
> E. Other issues
>
> •	Figures 6 and 7 have been updated with readable font sizes.
> •	Real data are taken as heterogeneous, whereas the synthetic data are regarded as homogeneous. High-dimensional data contain more features than samples as discussed in the Introduction of the manuscript.
> •	Consensus matrix here refers to the combined weight matrix as described in the manuscript. A clearer definition is now included in Section 2.3.
> •	The j-SNE citation has been fixed.
> •	The equations have been updated following your suggestion.
> •	A distance of infinity between samples not in the kNN graph is needed in order for the shortest path to be identified and the typo has been corrected.
> •	The description of ISOMAP has been updated to be more accurate.

---

### Review · Reviewer_xCvW · 2023-07-27

**Summary Of Contributions:**

This paper introduces extensions of manifold learning approaches in multi-view dataset, specifically named multi-SNE, multi-LL, and multi-ISOMAP. The proposed methodology modifies conventional algorithms by taking weighted combinations of each data-view. These algorithms show comprehensive projections to the low-dimensional space, compared to the ones obtained by visualizing each data-view separately. By employing Multi-SNE and other proposed algorithms, improvements are observed in terms of four metrics (Accuracy, Normalized Mutual Information, Rand Index, Adjusted Rand Index) as well as an enhanced 2-D visualization. The authors substantiate their claims through comprehensive experiments, providing qualitative results that demonstrate enhanced effectiveness on visualizing diverse real and synthetic datasets, including those derived from biological applications.

**Audience:**

Yes

**Broader Impact Concerns:**

There are no concerns regarding ethical implications of the work.

**Claims And Evidence:**

Yes

**Requested Changes:**

A major limitation of the paper is the lack of novelty on how it extends the existing algorithms. The concept of utilizing a weighted combination of each view does not seem to offer a significant impact. To address this concern effectively, a more extensive comparison between the proposed methodology and alternative approaches, including concatenation of data-views, m-SNE, m-LLE, among others, should be provided through a combination of experiments and theoretical analysis. This comprehensive evaluation is essential to substantiate the claim of novelty in the paper.

**Strengths And Weaknesses:**

The paper demonstrates several strengths, one of which is the straightforward extension of conventional visualization algorithms, enabling their application to other dimensionality reduction techniques for multi-view datasets. The proposed framework, comprising multi-SNE, multi-LLE, and multi-ISOMAP, extends these conventional algorithms by adopting a weighted average of low-dimensional embeddings for each data-view. The authors present intuitive scenarios supporting the viability of the multi-view approach (Figure 2) and provide visualization results, comparing the algorithms with single-view approaches (Figure 3, 4, 5).
However, the main weakness of the paper lies in its limited explanation of why the proposed algorithms outperform other dimensionality reduction approaches. Specifically, the authors need to clarify why treating each view separately yields better results compared to concatenating the views into a single set of data. Although Figure 3 appears to demonstrate the superiority of the proposed algorithm over the concatenation approach, the comparison is based on different hyperparameters, which hinder a fair assessment. To establish a solid foundation for the proposed algorithm's superiority over simple concatenation of multiple views, the authors should provide either experimental evidence or theoretical analysis.
Moreover, a comprehensive comparison with other existing methods is essential to highlight the novelty of the proposed methodologies. For instance, the proposed extensions, multi-SNE, and multi-LLE, seem similar to previous works such as m-SNE [1] and m-LLE [2]. To distinguish the proposed algorithms from these prior methods, the authors should clearly indicate the key differences and illustrate the novelty of their approach and its expected outcomes, preferably in theoretical analysis.
In conclusion, while the paper presents an extension of conventional visualization algorithms from various perspectives, there is a need for theoretical analysis or more robust experiments to justify the effectiveness of the proposed algorithm.

[1] Bo Xie, Yang Mu, Dacheng Tao, and Kaiqi Huang. m-sne: multiview stochastic neighbor embedding. IEEE Transactions on Systems, Man, and Cybernetics, Part B: Cybernetics, 41:1088–1096, 2011.

[2] Hualei Shen, Dacheng Tao, and Dianfu Ma. Multiview locally linear embedding for effective medical image retrieval. PLOS ONE, 8(12):1–21, 2013.

---

> ### Author Response · Authors · 2023-08-23
> **Comment 1 - Rebuttal**
>
> We would like to thank you for recognising our efforts for comparing our proposed methods with other multi-view methods proposed in the literature and with the equivalent single-view version of the three manifold learning approaches that we chose to explore in this manuscript: SNE, LLE and ISOMAP. Through our comprehensive comparison on four real datasets and three simulated datasets we have addressed the following main questions:
> - is multi-view visualisation better than single-view applied through data concatenation?
> - can we obtain robust clustering through multi-view visualisation?
> - should noisy data be discarded for optimising the obtained multi-view visualisations?
> - what is the effect of the multi-view hyper-parameter values?
>
> Through our conducted experiments we illustrated that multi-view visualisation outperforms simple concatenation of the data-views. This confirms previous literature results that argued that simple concatenation can lead to spurious findings (References).
>
> Through our experiments we identified multi-SNE to be the best performing approach, and we have subsequently further explored this approach by investigating a different weighting approach for the data-views, and an alternative multi-view pre-processing step based on CCA versus PCA.
>
> Even though in this manuscript we have mostly focused on the visualisations produced by our proposed methods, we have further illustrated that the projections of multi-SNE can be used as input data for obtaining robust clustering. Table 5 illustrates how multi-SNE combined with K-means can produce clusters that can outperform clustering obtained from multi-view clustering approaches proposed in the literature. In additional experiments presented in the Appendix, we further explored the effect of the clustering algorithm, by applying DBSCAN instead of K-means. From our analysis the two algorithms were found to have similar performance.
>
> We agree that a theoretical analysis illustrating the advantage of multi-SNE over m-SNE would have been beneficial. Unfortunately, due to the nature of the problem this is not straightforward as we illustrate in Comment 2.
>
> Ideally the best performing approach is the one with the smallest cost function. We have therefore computed empirically the cost functions of multi-SNE and m-SNE on the analysed data.  In all datasets, multi-SNE produced the lowest cost function of the two algorithms (Table 1). This further supports the superiority of multi-SNE versus m-SNE in the presented visualisations. A similar statement also holds for LLE and ISOMAP. As an illustrative example, the values of the cost functions of multi-LLE and m-LLE are presented (Table 2).
>
> We strongly believe that these empirical results further support the findings of our conducted study that show that multi-SNE can provide robust lower dimensional projections of the multi-view data that can be used for both visualisation and clustering.
>
> Table 1: Multi-SNE (C_multiSNE) and m-SNE (C_mSNE) cost function evaluation on the datasets analysed:
>
> - MMDS: C_multiSNE=1.96; C_mSNE=3.42
> - NDS: C_multiSNE=2.26; C_mSNE=3.83
> - MCS: C_multiSNE= 2.01; C_mSNE=3.37
> - Cancer types: C_multiSNE= 1.59; C_mSNE=3.38
> - Handwritten digits: C_multiSNE=1.84; C_mSNE=5.69
> - Caltech7: C_multiSNE= 1.97; C_mSNE=5.08
> - Caltech7 (balanced): C_multiSNE=1.40; C_mSNE=3.73
>
>
> Table 2: Multi-LLE (C_multiLLE) and m-LLE (C_mLLE) cost function evaluation on the real datasets analysed:
>
> - Cancer types: C_multiLLE=0.0296; C_mLLE=0.273
> - Handwritten digits: C_multiLLE=0.707; C_mLLE=0.906
> - Caltech7: C_multiLLE=0.161; C_mLLE=0.345
> - Caltech7 (balanced): C_multiLLE=0.162; C_mLLE=0.344

---

> ### Author Response · Authors · 2023-08-23
> **Comment 2 - Theoretical comparison and experimental evidence between multi-SNE and m-SNE**
>
> As previously discussed, the algorithm with the smallest cost function is the preferred one. Below we investigate in detail the cost functions of $C_{mSNE}$ and $C_{multiSNE}$ and present the number of factors that their evaluation depends on.
>
> The cost function of m-SNE is:
>
> $C_{mSNE} = \sum_i \sum_j (\sum_m \beta^m p_{ij}^m) \log \frac{\sum_m \beta^m p_{ij}^m}{\tilde{q_{ij}}} $
>
> The cost function of multi-SNE is:
>
> $C_{multiSNE} = \sum_i \sum_j \sum_m w^m p^m_{ij} \log \frac{p^m_{ij}}{q_{ij}} $
>
> where $\tilde{q_{ij}}$ and $q_{ij}$ represent the probability that samples $i$ and $j$ are neighbours in the low-dimensional space for m-SNE and multi-SNE, respectively. \\
>
> For simplicity we can assume that $w^m = \beta^m, \forall m$. Subtracting the two cost functions gives us:
>
> $C_{multiSNE} - C_{mSNE} =  \sum_i \sum_j \sum_m \beta^m p^m_{ij} \log \frac{p^m_{ij}}{q_{ij}} - \beta^m p_{ij}^m \log \frac{\sum_m \beta^m p_{ij}^m}{\tilde{q_{ij}}}$
>
> $=  \sum_{i \neq j} \sum_m p^m_{ij} \left[  \beta^m \log \frac{p^m_{ij}}{q_{ij}} - \beta^m \log \frac{\sum_m \beta^m p_{ij}^m}{\tilde{q_{ij}}} \right] $
>
> $=  \sum_{i \neq j} \sum_m p^m_{ij} \left[ \log \frac{p^m_{ij}}{q_{ij}} - \log \frac{\sum_m \beta^m p_{ij}^m}{\tilde{q_{ij}}} \right]$
>
> $=  \sum_{i \neq j} \sum_m p^m_{ij} \left[ \log \frac{\frac{p^m_{ij}}{q_{ij}}}{\frac{\sum_m \beta^m p_{ij}^m}{\tilde{q_{ij}}}} \right] $
>
> $=  \sum_{i \neq j} \sum_m p^m_{ij} \left[ \log \frac{p^m_{ij}}{\sum_m \beta^m p_{ij}^m} + \log \frac{\tilde{q_{ij}}}{q_{ij}} \right]$
>
> $=  \sum_{i \neq j} \sum_m KL(P^{(m)} || \hat{P}) + \sum_{i \neq j} \sum_m p^m_{ij} \log \frac{\tilde{q_{ij}}}{q_{ij}}  $
>
> where $\hat{P} = \sum_m \beta^m P^{(m)}$ and $\sum_m \beta^m = 1$. Since $ KL(X||Y) \geq 0$ for any probability distributions $X, Y$, the first part of the equation is positive. The second part of the equation depends on the probability distributions of the corresponding low-dimensional embeddings ($\tilde{q_{ij}}$ for m-SNE and $q_{ij}$ for multi-SNE). In the unrealistic situation where the two projections are the same then the second term is equal to zero, and therefore $C_{multiSNE}>C_{mSNE}$. But this situation is very unlikely to happen.
>
>
> Quantifying $C_{multiSNE}-C_{mSNE}$ depends on a number of unknown factors that it is not straightforward to compute for general cases. These factors include the probability distribution of the original data and the KL divergence between the probability distributions of the original data and the constructed projections.

---

### Review · Reviewer_kbGJ · 2023-08-20

**Summary Of Contributions:**

This submission proposes variations of manifold learning algorithms, namely, SNE, LLE and ISOMAP, in the context of multiview learning. It introduces a weighting factor and combines the loss function of each view using a weighted sum. The manuscript also presents a series of numerical results to demonstrate the effectiveness of the proposed approach, including feeding the manifold learning results to K-means and applying the learned visualization to biological data.

**Audience:**

No

**Broader Impact Concerns:**

This submission seems to have not had a section on broader impacts.

**Claims And Evidence:**

No

**Requested Changes:**

Some suggestions:

1) It would be nice to see the reasons why using the proposed weighted combinations can lead to better manifold learning results. Other than merely proposing a modified loss, the society would benefit more if the insights behind the proposed losses can be clearly stated.

2) It would also be nice to have some theoretical justifications, if possible. Considering all the modified methods are all well known and well studied. Additional understanding in theory would definitely add intellectual merit to the existing knowledge.

3) The empirical results can be enriched by considering larger, imbalanced, and more recent datasets. Baselines can use some modern foundation model trained embeddings (e.g., CLIP + t-SNE).

**Strengths And Weaknesses:**

Some strengths are noted by the reviewer:

1) The submission has a clear statement of the problem and also clear expression of the existing manifold learning algorithms.

2) The submission is quite easy to follow and read. The writing is clear and smooth.

Some critical weakness:

1) The proposed approach seems to be too straightforward, if not trivial. The proposed method is simply combining the existing t-SNE, LLE and ISOMAP losses of different views using a weighted sum. The weight is set to be w_m = 1/M across this paper in most of the part, where M is the number of views.  This does not seem to constitute a significant technical contribution. Easy-to-implement approaches are indeed appreciated, but such simple approaches should be supported by good rationale or theoretical analysis to be convincing.

2) The technical insight that could be gained from reading this article seems to be on the limited side. Other than proposing such a weighted combination-type variation of existing losses, the manuscript does not have much novelty to be noted. The rationale and motivation of using the proposed method and the choice of w_m were not discussed. There are some procedures of adjusting the weights in a later part of the paper, but it was unclear how this adjustment would affect the algorithm's convergence.

3) The experiments seem to be not very convincing. The synthetic data only had 3-5 clusters and the cluster sizes are balanced. This does not poses a lot of challenges to dimensionality reduction problems. The real data are also classic small size datasets that do not represent new challenges in multiview learning in NLP or computer vision.

---

> ### Author Response · Authors · 2023-08-23
> **Comment 1 - Rebuttal**
>
> We would like to thank you for your comments and recommendations for improving our manuscript. Below we have addressed the raised issues. We hope that you will find our updated manuscript in response to your comments suitable for publication.
>
> A. Theoretical justification
>
> Reviewer xCvW expressed a similar concern and our response on  this matter is the same.
>
> We agree that a theoretical analysis illustrating the advantage of multi-SNE over m-SNE would have been beneficial. Unfortunately, due to the nature of the problem this is not straightforward as we illustrate in our second comment.
>
> Ideally the best performing approach is the one with the smallest cost function. We have therefore computed empirically the cost functions of multi-SNE and m-SNE on the analysed data.  In all datasets, multi-SNE produced the lowest cost function of the two algorithms (Table 1). This further supports the superiority of multi-SNE versus m-SNE in the presented visualisations. A similar statement also holds for LLE and ISOMAP. As an illustrative example, the values of the cost functions of multi-LLE and m-LLE are presented (Table 2).
>
> Table 1: Multi-SNE (C_multiSNE) and m-SNE (C_mSNE) cost function evaluation on the datasets analysed:
>
> - MMDS: C_multiSNE=1.96; C_mSNE=3.42
> - NDS: C_multiSNE=2.26; C_mSNE=3.83
> - MCS: C_multiSNE= 2.01; C_mSNE=3.37
> - Cancer types: C_multiSNE= 1.59; C_mSNE=3.38
> - Handwritten digits: C_multiSNE=1.84; C_mSNE=5.69
> - Caltech7: C_multiSNE= 1.97; C_mSNE=5.08
> - Caltech7 (balanced): C_multiSNE=1.40; C_mSNE=3.73
>
>
> Table 2: Multi-LLE (C_multiLLE) and m-LLE (C_mLLE) cost function evaluation on the real datasets analysed:
>
> - Cancer types: C_multiLLE=0.0296; C_mLLE=0.273
> - Handwritten digits: C_multiLLE=0.707; C_mLLE=0.906
> - Caltech7: C_multiLLE=0.161; C_mLLE=0.345
> - Caltech7 (balanced): C_multiLLE=0.162; C_mLLE=0.344
>
> B. Analysis of larger, imbalanced, more recent datasets
>
> Our manuscript covers a range of different characteristics of datasets, for different number of clusters, features, data-views along with heterogeneous and homogeneous datasets.
>
> The synthetic datasets were simulated to contain balanced number of features for simplicity, since the aim of their analysis was to validate the algorithms, while the analysis of real datasets (with imbalanced number of features) showcase the application of our proposed methods. In particular, cancer types dataset contains two large data-views (10299 and 22503 features).
>
> Further, we have applied multi-SNE on scRNA-seq multi-modal data, which are a challenging and modern data source, and application area. We believe that our manuscript contains extensive and comprehensive comparisons and applications of the proposed methods, highlighting their robustness and limitations.
>
> We believe that the exploration of modern foundation model trained embeddings such as CLIP + t-SNE, as it was proposed, is beyond the original scope of the paper that explores the extension of existing manifold approaches to multi-view data. We attempted to cover most possible cases and we have extended classical PCA-based t-SNE with multi-CCA as an initial step on the challenging real datasets.
>
> C. Weights
>
> The scope of our manuscript is to introduce and propose new multi-view manifold learning algorithms that can represent visually the global underlying truth of all available data-views, even when single-view algorithms produce conflicting visualisations. Thus, we did not focus on the adjustment of weights for each data-view. If our proposed algorithm is found to be superior to single-view solutions without optimal weights, then it should perform better with optimal weights.
>
> In agreement with the reviewer, it is an important part of the algorithms, and it should be addressed in further detail through additional analyses and solutions to identify the optimal weights. Such additional explorations would be beyond the scope of our manuscript which aims to highlight the importance of using multi-view solutions over single-view ones when applicable. As identified by the reviewer, we outlined a recent approach in adjusting the weights as a variation that could be introduced on top of our proposals, and it would require further extensive comparisons to assess its validity and performance.

---

> ### Author Response · Authors · 2023-08-23
> **Comment 2 - Theoretical comparison and experimental evidence between multi-SNE and m-SNE**
>
> As previously discussed, the algorithm with the smallest cost function is the preferred one. Below we investigate in detail the cost functions of $C_{mSNE}$ and $C_{multiSNE}$ and present the number of factors that their evaluation depends on.
>
> The cost function of m-SNE is:
>
> $C_{mSNE} = \sum_i \sum_j (\sum_m \beta^m p_{ij}^m) \log \frac{\sum_m \beta^m p_{ij}^m}{\tilde{q_{ij}}} $
>
> The cost function of multi-SNE is:
>
> $C_{multiSNE} = \sum_i \sum_j \sum_m w^m p^m_{ij} \log \frac{p^m_{ij}}{q_{ij}} $
>
> where $\tilde{q_{ij}}$ and $q_{ij}$ represent the probability that samples $i$ and $j$ are neighbours in the low-dimensional space for m-SNE and multi-SNE, respectively. \\
>
> For simplicity we can assume that $w^m = \beta^m, \forall m$. Subtracting the two cost functions gives us:
>
> $C_{multiSNE} - C_{mSNE} =  \sum_i \sum_j \sum_m \beta^m p^m_{ij} \log \frac{p^m_{ij}}{q_{ij}} - \beta^m p_{ij}^m \log \frac{\sum_m \beta^m p_{ij}^m}{\tilde{q_{ij}}}$
>
> $=  \sum_{i \neq j} \sum_m p^m_{ij} \left[  \beta^m \log \frac{p^m_{ij}}{q_{ij}} - \beta^m \log \frac{\sum_m \beta^m p_{ij}^m}{\tilde{q_{ij}}} \right] $
>
> $=  \sum_{i \neq j} \sum_m p^m_{ij} \left[ \log \frac{p^m_{ij}}{q_{ij}} - \log \frac{\sum_m \beta^m p_{ij}^m}{\tilde{q_{ij}}} \right]$
>
> $=  \sum_{i \neq j} \sum_m p^m_{ij} \left[ \log \frac{\frac{p^m_{ij}}{q_{ij}}}{\frac{\sum_m \beta^m p_{ij}^m}{\tilde{q_{ij}}}} \right] $
>
> $=  \sum_{i \neq j} \sum_m p^m_{ij} \left[ \log \frac{p^m_{ij}}{\sum_m \beta^m p_{ij}^m} + \log \frac{\tilde{q_{ij}}}{q_{ij}} \right]$
>
> $=  \sum_{i \neq j} \sum_m KL(P^{(m)} || \hat{P}) + \sum_{i \neq j} \sum_m p^m_{ij} \log \frac{\tilde{q_{ij}}}{q_{ij}}  $
>
> where $\hat{P} = \sum_m \beta^m P^{(m)}$ and $\sum_m \beta^m = 1$. Since $ KL(X||Y) \geq 0$ for any probability distributions $X, Y$, the first part of the equation is positive. The second part of the equation depends on the probability distributions of the corresponding low-dimensional embeddings ($\tilde{q_{ij}}$ for m-SNE and $q_{ij}$ for multi-SNE). In the unrealistic situation where the two projections are the same then the second term is equal to zero, and therefore $C_{multiSNE}>C_{mSNE}$. But this situation is very unlikely to happen.
>
>
> Quantifying $C_{multiSNE}-C_{mSNE}$ depends on a number of unknown factors that it is not straightforward to compute for general cases. These factors include the probability distribution of the original data and the KL divergence between the probability distributions of the original data and the constructed projections.

---

### Author Response · Authors · 2023-08-23
**Rebuttal letter**

Dear Editors and Reviewers,

We would like to thank the action editor, Professor Seungjin Choi and the three anonymous reviewers for their evaluation, comments and recommendations that indisputably have improved our manuscript.

As Reviewer gKV1 has pointed out our paper addresses a significant problem involving the visualisation, and clustering, of multi-view data. This is a major challenge in many application fields that collect multi-view data including single-cell biology. An example of single-cell multi-view data has been included in the manuscript to illustrate the applicability of our developed methods to the field.

Reviewer xCvW enquired why treating each data view separately yields better results compared to concatenating the views into a single set of data. Firstly to clarify our proposed methods do not treat each data view separately but instead take weighted combinations of the available data-views. As discussed in the literature and illustrated in our manuscript due to the different data characteristics concatenating the views can lead to spurious results. For our conducted analyses we have taken all measures possible to optimise each approach (whether single-view or multi-view). This includes identifying the optimal hyper-parameter of each method/data-view for each one of the visualisations produced. This is the reason why in Figure 3 each visualisation is based on a different hyper-parameter. We disagree with the reviewer xCvW that this hinders our assessment, as imposing the same hyper-parameter would have instead given an unfair assessment. The optimal hyper-parameter was found through a comprehensive search from a range of possible hyper-parameter values performed for each method/data-view. In addition, before concatenating the data views we first normalised them. This was part of our conducted analysis throughout. This was something that we missed including in the manuscript beforehand, but we have now updated the manuscript to include it.

Reviewer kbGJ expressed some concerns with regard to theoretical justification and our choice of keeping the weights constant among the data-views ($w_m = \frac{1}{M}$). The scope of our manuscript is to introduce and propose new multi-view manifold learning algorithms that are capable of representing visually the global underlying truth of all available data-views, even when single-view algorithms produce conflicting visualisations. As identified by the reviewer, we outlined a recent approach in adjusting the weights as a variation that could be introduced on top of our proposals. It would require further extensive comparisons to assess its validity and performance. The synthetic datasets were chosen to contain a balanced number of features for simplicity, since the aim of the simulations was to validate the algorithm, while the real datasets (with an imbalanced number of features) showcase the implementation of our proposed methods, including their application on multi-modal scRNA-seq data, a challenging and modern data source.

We appreciate Reviewer's gKV1 comments about setting the stone straight with regard to the parallel publication. As they have correctly pointed out both manuscripts first appeared online as preprints in January 2021 with a week difference. We have attempted to make this clearer to the readership by adding more information regarding this in the Introduction of the manuscript. We have further addressed the reviewer's concerns about the description of the methods and simulation study by adding more information and details in the manuscript.

Our manuscript includes a comprehensive comparison between our proposed approaches: multi-SNE, multi-LLE and multi-ISOMAP with other multi-view dimensionality approaches based on these approaches including, m-SNE and m-LLE. In addition, the single-view version of SNE, LLE and ISOMAP was applied to the concatenated version of the data. The approaches were compared on both simulated and real datasets with different characteristics, including different noise levels, dimensions, number of clusters, etc. Our manuscript provides experimental evidence of the superiority of our proposed algorithm by exploring analytically various scenarios with different data characteristics to showcase the robustness of the proposed solutions.

By addressing the comments of the reviewers we have updated the manuscript, and all the changes made are highlighted in {\color{red} red}. We hope that you will find the updated version of our manuscript suitable for publication in the Transactions on Machine Learning Research.

Yours sincerely,
Theodoulos Rodosthenous,
on behalf of the authors

---

### Decision · Action_Editors · 2023-09-29

**Recommendation:** Reject

**Comment:**

All reviewers feel that the authors have done a good job in their rebuttal, resolving most of issues raised by reviewers.  The main idea is to consider a weighted sum of loss of each view for multi-view manifold learning. The method in this paper is a straightforward extension of existing algorithms to multi-view domains, so it does not bring much insights and ideas that can be of interest for TMLR audience.
Moreover it is criticized that the claim on "using multiple views benefits visualization" is too vague since the paper does not have much insight revealed regarding this claim.  Therefore, the paper is not recommended for acceptance in its current form. I hope authors found the review comments informative and can improve their paper by addressing these carefully in future submissions to other venue.

**Audience:**

Multi-view data are available in many applications. It becomes more imperative to handle multi-view data in machine learning community. On the other hand, data visualization methods such as t-SNE and UMAP are frequently used to visualize the high-dimensional outcomes in a low-dimensional space. Thus, the topic itself in this paper is timely and interesting.
However, a similar idea was already published in Genome Biology in 2021 although two papers (one for the arxiv version of the current work and the other in Genome Biology) initially appeared almost simultaneously on arXiv. Since they have been available since 2021, the current work is unlikely to be of interest to the TMLR audience now, without new insights compared to 2021 papers.

**Claims And Evidence:**

This paper presents an extension of existing manifold learning algorithms (SNE, LLE, Isomap) to the multi-view environment. The main idea is very simple, introducing a weighted sum of loss of each view to determine a low-dimensional manifold of multi-view data. Numerical experiments on a few different datasets demonstrated the validity and usefulness of the proposed methods.